# Parabrachial CGRP neurons modulate active defensive behavior under a naturalistic threat

Gyeong Hee Pyeon, Hyewon Cho, Byung Min Chung, June-Seek Choi, Yong Sang Jo*

School of Psychology, Korea University, Seoul, Republic of Korea

**Abstract** Recent studies suggest that calcitonin gene-related peptide (CGRP) neurons in the parabrachial nucleus (PBN) represent aversive information and signal a general alarm to the forebrain. If CGRP neurons serve as a true general alarm, their activation would modulate both passive and active defensive behaviors depending on the magnitude and context of the threat. However, most prior research has focused on the role of CGRP neurons in passive freezing responses, with limited exploration of their involvement in active defensive behaviors. To address this, we examined the role of CGRP neurons in active defensive behavior using a predator-like robot programmed to chase mice. Our electrophysiological results revealed that CGRP neurons encode the intensity of aversive stimuli through variations in firing durations and amplitudes. Optogenetic activation of CGRP neurons during robot chasing elevated flight responses in both conditioning and retention tests, presumably by amplifying the perception of the threat as more imminent and dangerous. In contrast, animals with inactivated CGRP neurons exhibited reduced flight responses, even when the robot was programmed to appear highly threatening during conditioning. These findings expand the understanding of CGRP neurons in the PBN as a critical alarm system, capable of dynamically regulating active defensive behaviors by amplifying threat perception, and ensuring adaptive responses to varying levels of danger.

*For correspondence:
ysjo@korea.ac.kr

Competing interest: The authors declare that no competing interests exist.

## Editor's evaluation

This valuable work advances our understanding of parabrachial CGRP threat function. The evidence supporting CGRP aversive outcome signaling to promote active defensive behavior is solid. The work will be of interest to neuroscientists studying defensive behaviors.

## Introduction

Effective survival necessitates a repertoire of dynamic defensive behaviors, encompassing both passive and active responses. Passive defensive strategies, such as freezing, help avoid detection from predators by reducing motion (*Blanchard and Blanchard, 1969a*; *Fanselow, 1980*; *Fanselow, 1982*). In contrast, active defensive behaviors, including fleeing or fighting, enable animals to swiftly escape or confront imminent threats (*Blanchard and Blanchard, 1969b*; *Bolles, 1970*). The ability to adaptively switch between passive and active defenses in response to varying threat contexts is essential for optimizing survival outcomes, as demonstrated by studies utilizing naturalistic threat stimuli like predator-like robots or looming disks, which allowed the observation of various critical defensive behaviors (*Choi and Kim, 2010*; *Kang et al., 2022*; *Pyeon et al., 2023*; *Telensky et al., 2011*). A critical component of this adaptive response is the general alarm signal, which detects danger and plays a role in eliciting appropriate defensive behaviors in the face of threats. These signals help organisms

**eLife digest** How animals decide to respond to threatening situations can be the difference between life and death. Most animals show different defensive behaviors depending on how severe the threat is. For example, if it is imminent, they may flee to escape. On the other hand, if a threat is less severe, they may freeze to avoid detection.

The brain uses its own 'general alarm system' to help recognize and respond to threats. This system is made up of nerve cells which detect potential threat signals, 'analyze' them, and relay information about them to other parts of the brain that trigger the appropriate response. In humans, imbalances in this response can lead to maladaptive defense responses, such as those seen in anxiety or post-traumatic stress disorders, where fear and avoidance responses are excessive in relation to the threat.

One population of nerve cells, known as CGRP neurons, can detect a wide range of signals, and are known to respond by triggering passive behavior, such as freezing. However, whether CGRP neurons also trigger active behaviors, such as fleeing, remained unclear. Therefore, Pyeon et al. set out to study how CGRP neurons influence both passive and active defensive behaviors in response to varying threat levels.

To create realistic 'models' of different threat intensities in the laboratory, Pyeon et al. used a predator-like robot programmed to chase mice at different speeds. The mice were genetically modified so that researchers could record the activity of CGRP neurons, as well as activate the neurons artificially using light.

Activating CGRP neurons in mice being chased at a slow speed led to fleeing responses comparable to those observed during a higher-speed chase. This suggests that enhancing the alarm signal by artificially activating CGRP neurons may have caused the mice to perceive the threat as more intense and to react as though the danger was greater than it actually was. In contrast, mice with their CGRP neurons artificially 'switched off' were very unlikely to flee or freeze regardless of the speed of the chase.

The findings reveal that CGRP neurons respond differently to varying threat levels and regulate both passive and active defensive behaviors. This highlights their important role in adapting defensive responses to the severity of the threat. Building on these insights, future studies could explore strategies to regulate CGRP neuron activity, potentially leading to therapeutic approaches to address conditions marked by exaggerated or insufficient threat responses.

quickly recognize and respond to potential threats. The mechanisms underlying these alarm signals can be studied through Pavlovian fear conditioning (*Bolles and Collier, 1976*; *Fanselow and Poulos, 2005*; *LeDoux, 2000*; *Maren, 2001*). In this process, a neutral sensory stimulus (conditioned stimulus or CS) is paired with an aversive unconditioned stimulus (US), leading to a conditioned response (CR) that can be expressed as either freezing or fleeing, depending on the specific features of the CS (*Borkar and Fadok, 2021*; *Fadok et al., 2017*) and US (*Lee et al., 2018*; *Pyeon et al., 2023*).

Neurons within the PBN that express CGRP have been suggested to function as general alarm signals in the brain (*Palmiter, 2018*). These neurons respond to noxious stimuli of diverse sensory modalities (*Campos et al., 2018*; *Carter et al., 2013*; *Chen et al., 2018*; *Kang et al., 2022*) and transmit interoceptive and exteroceptive information to the forebrain (*Bernard and Besson, 1988*; *Chiang et al., 2019*). Additionally, these CGRP neurons relay US information to the central amygdala during conventional fear conditioning with electric footshock (*Han et al., 2015*). Prior studies have primarily focused on the role of CGRP neurons in mediating passive freezing behavior, demonstrating that activation of these neurons exclusively elicits immediate freezing behavior and contributes to the formation of fear memories (*Bowen et al., 2020*; *Han et al., 2015*). However, for CGRP neurons to serve as a true general alarm system, they must be capable of transmitting threat-related signals and facilitating the coordination of appropriate defensive behaviors, whether passive or active. While the role of CGRP neurons in passive responses is well-established, their potential involvement in active defensive behaviors remains unexplored.

To address this, we employed a more dynamic and ecologically relevant US by using a predator-like robot to chase the animals, thereby incorporating an imminent threat. We hypothesized that CGRP neurons modulate adaptive defensive behaviors depending on the severity or type of threat.

We first recorded CGRP neuron activity in response to various aversive stimuli including the robot chasing to determine whether they encode noxious stimuli differentially. We then manipulated CGRP activity—both activating and inactivating—during fear conditioning with robot chasing and foot-shock. Our results suggest that manipulation of CGRP neurons bidirectionally modulates conditioned fleeing behaviors through altering the perception of the threat. These results highlight the role of CGRP neurons as a general alarm signal, primarily facilitating passive defensive behaviors, while also engaging in active defensive behaviors in response to high-threat conditions.

## Results

### Differential responses of CGRP neurons to aversive stimuli of varying intensities

The response profiles of CGRP neurons in conventional fear conditioning with footshock have been well-reported (*Han et al., 2015*). However, how CGRP neurons respond to chasing threats has not been established. To investigate the activity of CGRP neurons in response to robot chasing, in vivo recordings using the optical-tagging strategy were performed (*Jo et al., 2018*; *Juarez et al., 2023*). Heterozygous mice expressing Cre-recombinase at the *Calca* locus (*Calca^{Cre/+}*) were injected with a Cre-dependent adeno-associated virus (AAV) carrying an excitatory channelrhodopsin (ChR2) with red fluorescent protein (AAV-DIO-ChR2-mScarlet). Then, a movable optrode array containing one optic fiber with four tetrodes was implanted over the PBN (*Figure 1A*).

After 2 wk of recovery, neuronal activity was recorded during fear conditioning with a robot over three consecutive days (*Figure 1B*). Animals were first habituated to a tone (4 kHz; 70 dB; 10 s) as a CS in a rectangular box. The following day, the animals were placed in a donut-shaped maze and presented with the CS 10 times, each paired with an US of being chased by the robot at a speed of 70 cm/s for 3 s. When animals collided with the robot, it pushed them forward, increasing their fleeing speed (*Figure 1—figure supplement 1C*). If an animal blocked its path, the robot continued to push it, but its speed decreased due to friction and reduced motor power, ensuring it did not run over the animal. On day 3, fear memory was assessed by presenting the CS alone 10 times in the same context as the habituation session. In this behavioral paradigm, animals engaged in both passive and active defensive strategies, as evidenced by freezing and flight responses during conditioning (*Figure 1—figure supplement 1C*). Passive behavior was measured by freezing, defined as the absence of movement, while active behavior was represented by flight, quantified with a flight score calculated by dividing the average velocity during the CS by the average velocity during the pre-CS period (*Borkar et al., 2024*; *Fadok et al., 2017*). While both responses were observed during conditioning, the test day, conducted without the robot, showed increased freezing and reduced fleeing responses (*Figure 1C and D*).

To identify CGRP neurons, 10 pulses of blue light (5 ms duration) at 30 Hz were delivered 10 times at the end of each behavioral recording session. Out of 183 PBN neurons, 84 cells with a high probability of light-evoked spikes (>0.8) and a short spike latency (<5.5 ms) after light onset were classified as CGRP neurons (*Figure 1E*). Compared to habituation, CGRP neurons showed significantly increased excitation to the CS during conditioning and retention, but only within the first 1 s after CS onset (1.5-fold increase); this difference became non-significant starting at 2 s (*Figure 1G*). However, these neurons exhibited significant excitation to the US with a fourfold increase (*Figure 1F and G*). Our findings using the robot as the US revealed that CGRP neurons primarily represent US information, albeit to a lesser extent, the onset of US-predictive information.

Given that CGRP neurons preferentially respond to the aversive US, we next asked how CGRP neurons encode different types of aversive stimuli. To address this, we monitored the activity of CGRP neurons while the animals received three types of stimuli, each varying in perceived threat intensity: (1) a pinprick to the hind paw using a needle (approximately 0.5 s); (2) a tail pinch 2 cm from the tail base using forceps (1 s); and (3) being chased by a robot (3 s). These aversive stimuli elicited different defensive behaviors. The pinprick caused hind paw withdrawal, and the tail pinch triggered vocalizations (audible squeaks) and immediate escaping behavior, indicating the highest threat intensity, whereas the robot chasing prompted only escaping behavior without any vocalization. CGRP neurons showed significantly excited firing that was time-locked to the onset of all three aversive stimuli and maintained this activity throughout the duration of each stimulus. After the offset of the

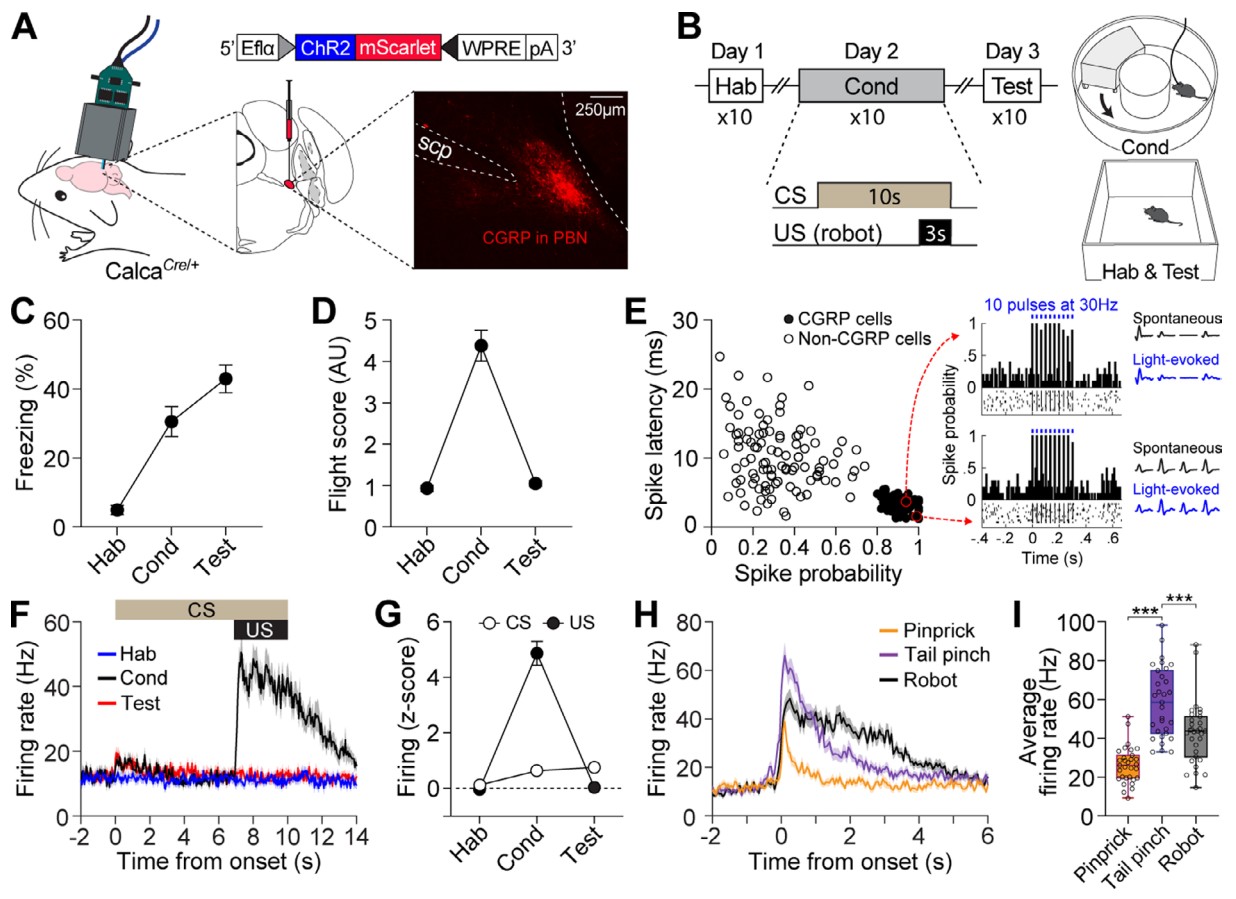

**Figure 1.** Distinct firing response patterns of calcitonin gene-related peptide (CGRP) neurons to different aversive stimuli. (**A**) Schematic of AAV-DIO-ChR2-mScarlet injection and optrode implantation into the parabrachial nucleus (PBN) of *Calca^{Cre/+}* mice (n=6) and the corresponding representative histology image. (**B**) Procedures for fear conditioning experiments with the chasing robot and a schematic diagram of the context used. (**C–D**) Freezing (**C**) and fleeing behaviors (**D**) in response to the conditioned stimulus (CS) during habituation, conditioning, and retention test. (**E**) Characteristics of light-evoked responses. Neurons with a short spike latency and a high spike probability response to light stimulation (filled circles) were classified as CGRP neurons. Inset: histograms showing firing patterns of two representative opto-tagged CGRP neurons response to 10 blue light pulses at 30 Hz. (**F**) Population firing rates of all recorded CGRP neurons (hab: n=28; cond: n=29; test: n=27) during fear conditioning with the robot. (**G**) Normalized firing in response to CS and unconditioned stimulus (US). (**H**) Population responses of all recorded CGRP neurons (n=31) in response to three aversive stimuli. (**I**) Average firing rates of CGRP neurons to pinprick, tail pinch, and robot chasing. Among these aversive stimuli, tail pinch induced a significantly greater increase in firing rates compared to both pinprick and robot chasing (one-way ANOVA, F(2, 90)=35.87, p<0.001; post-hoc tests, <i>p-values <0.001). ***p<0.001.

The online version of this article includes the following source data and figure supplement(s) for figure 1:

**Source data 1.** Behavior data for *Figure 1*.

**Source data 2.** Electrophysiology data for *Figure 1*.

**Figure supplement 1.** Characterization of CGRP neuron activity and behavioral responses during fear conditioning with chasing robot.

**Figure supplement 1—source data 1.** Correlations between spontaneous and light-evoked waveforms.

**Figure supplement 1—source data 2.** Velocity data for a representative animal across 10 fear conditioning trials.

stimuli, neuronal activity gradually restored back to the baseline with a slight delay (*Figure 1H*). These aversive stimuli elicited significantly different amplitudes of firing rates (*Figure 1I*). During the tail pinch, which generated the strongest defensive behavior, CGRP neurons exhibited the highest excitation amplitude, ranging from 33 to 98 Hz, with an average of 58 Hz. Taken together, these results indicate that CGRP neurons represent the temporal characteristics and intensity of different aversive stimuli through variations in firing duration and amplitude.

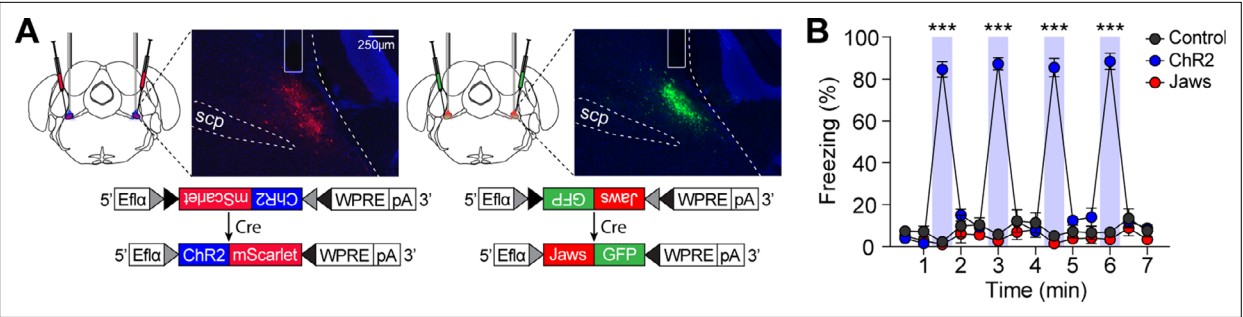

**Figure 2.** Stimulation of calcitonin gene-related peptide (CGRP) neurons in the absence of any external stimuli induces robust freezing behavior. (**A**) Schematic of bilateral AAV-DIO-ChR2-mScarlet or AAV-DIO-Jaws-GFP injections and optic fiber implantation into the parabrachial nucleus (PBN), with representative histological images of viral expression. (**B**) 30 s stimulation of CGRP neurons at 40 Hz resulted in significantly higher time-locked freezing behaviors in the ChR2 group compared to both the Jaws and control groups (n=10 per group; significant group ×time interaction in a repeated-measures two-way ANOVA, F(26, 351)=61.32, p<0.001; post-hoc tests at each time, *p*-values <0.001. ***p<0.001.

The online version of this article includes the following source data for figure 2:

**Source data 1.** Behavior data for *Figure 2*.

## CGRP activation promotes conditioned fleeing in robot chasing and conditioned freezing in footshock

To determine whether increasing or decreasing CGRP neuronal activity would induce defensive behaviors other than freezing, we first observed which defensive behaviors were elicited by either stimulating or inhibiting CGRP neurons in the absence of any external stimuli. *Calca*$^{Cre/+}$ mice were randomly assigned to groups and bilaterally injected with either AAV-DIO-ChR2-mScarlet for activation, AAV-DIO-Jaws-GFP for inactivation, or AAV-DIO-eYFP for control, followed by the implantation of optic fibers over the PBN (*Figure 2A*). To activate CGRP neurons, mice received 30 s of 40 Hz photostimulation, delivered four times, based on the observed spontaneous firing rate of approximately 43 Hz in response to robot chasing (*Figure 1I*). For inactivation, CGRP neurons were inhibited for 2 s, followed by a 1 s ramp down, repeated in cycles until a total duration of 30 s. Jaws and control groups showed no difference in movement during light on and off phases, indicating that light delivery did not alter their defensive behavior (*Figure 2B*). However, activation of CGRP neurons immediately induced robust freezing behavior, consistent with previous studies (*Bowen et al., 2020*; *Han et al., 2015*). These results confirmed that stimulating CGRP neurons without external aversive stimuli generates rapid, unconditioned freezing behavior in mice.

We then tested whether manipulating the activity of CGRP neurons during fear conditioning with robot chasing promotes fleeing behavior or amplifies freezing behavior. To effectively enhance the general alarm signal, additional activation of CGRP neurons was applied at 30 Hz during the robot chasing. *Calca*$^{Cre/+}$ mice underwent the fear conditioning paradigm in which the CS was paired with the robot chasing (3 s, 70 cm/s), with CGRP neurons selectively activated (30 Hz) or inhibited (3 s on and 1 s ramp down) throughout the presentation of the chasing (*Figure 3A*). To rule out the possibility that the observed behavior was merely a reaction to the cue and confirm that it resulted from the CS-US pairing, an unpaired group was included. In this group, the CS was not paired with the robot chasing; instead, the robot chasing was delivered within the inter-trial interval. Physical bumping occurred during robot chasing, potentially influencing the perception of threat. To ensure that differences in defensive behavior were not due to variations in bumping among the four groups, we analyzed bumping incidents and found no significant differences (*Figure 3—figure supplement 1C*; *Video 1*). This confirms that any observed differences in defensive responses are likely attributable to alterations in CGRP neuronal activity.

During conditioning, all four groups demonstrated fear memory formation, as evidenced by a progressive increase in freezing levels (*Figure 3B*). However, the Jaws and unpaired groups showed significantly lower freezing levels than the control and ChR2 groups. For the flight score on the conditioning day, the ChR2 group displayed significantly higher levels compared to the other three groups (*Figure 3C*), while the Jaws and unpaired groups consistently had lower flight scores than the control

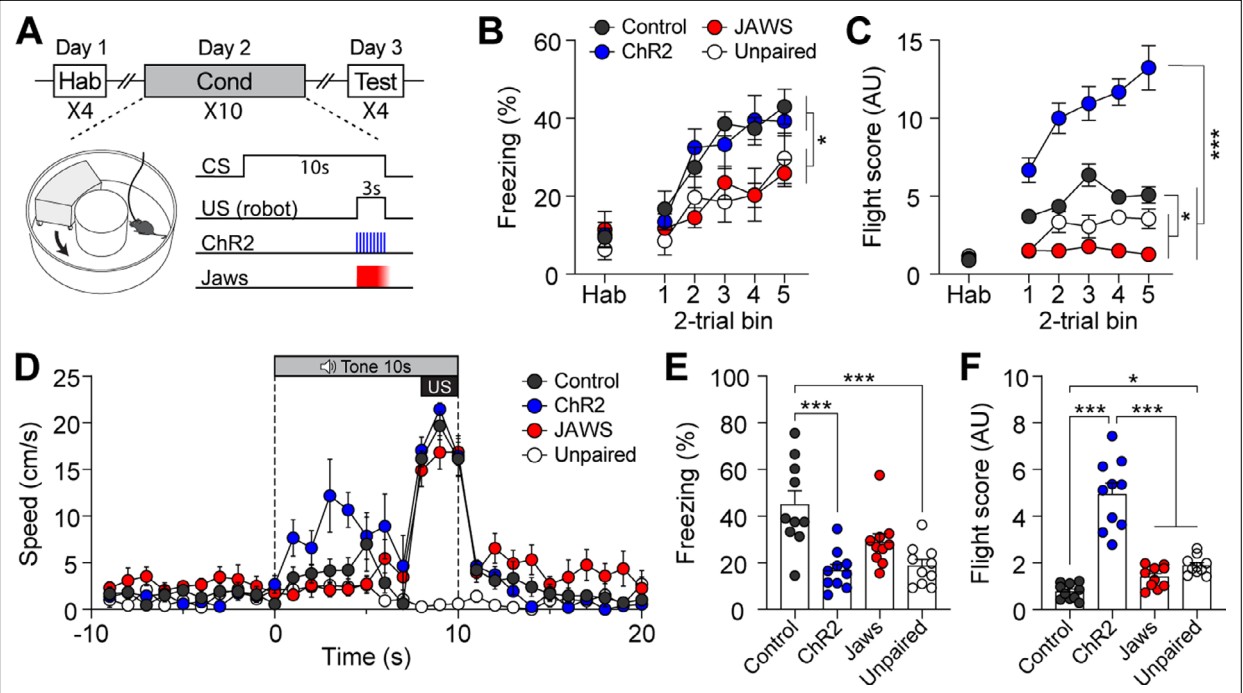

**Figure 3.** Activation of calcitonin gene-related peptide (CGRP) neurons during unconditioned stimulus (US) presentation enhances active defensive behavior. (**A**) A schematic diagram of fear conditioning protocol with the robot. CGRP neuronal activity was bidirectionally manipulated during the presentation of the robot chasing. (**B**) Freezing to the conditioned stimulus (CS) during habituation and conditioning sessions. A progressive increase in freezing was observed in all four groups (n=10 per group), but the Jaws and unpaired groups showed significantly lower freezing levels compared to the other two groups (significant group effect in a repeated-measures two-way ANOVA, F(3, 36)=5.50, p<0.01; subsequent post-hoc tests, <i>p-values <0.05). (**C**) Flight scores during habituation and conditioning sessions. The ChR2 group displayed significantly higher flight scores compared to other three groups (significant group effect in a repeated-measures two-way ANOVA, F(3, 36)=102.05, p<0.001; post-hoc tests, <i>p-values <0.001), while both the Jaws and unpaired groups had lower flight scores than the control group (<i>p-values <0.05). (**D**) Average velocities in response to the CS during the conditioning. The average velocity of the ChR2 group during the CS (first 7 s from onset) was significantly higher compared to that observed in all three groups (significant group effect in a repeated-measures two-way ANOVA, F(3, 36)=20.47, p<0.001; post-hoc test, <i>p-values <0.01). (**E**) Average freezing in response to the CS during the retention test. The ChR2 group froze significantly less than the control group (one-way ANOVA, F(3, 36)=10.82, p<0.001; post-hoc test, <i>p-value <0.001). The unpaired group also exhibited significantly lower freezing compared to the control group (<i>p-value <0.001). (**F**) Average flight scores in response to the CS during the retention test. The ChR2 group exhibited significantly higher flight scores than all three groups (one-way ANOVA, F(3, 36)=50.56, p<0.001; post-hoc test, <i>p-values <0.001). The unpaired group showed a significantly lower flight response compared to the control group (<i>p-value <0.05). *p<0.05, ***p<0.001.

The online version of this article includes the following source data and figure supplement(s) for figure 3:

**Source data 1.** Behavior data for *Figure 3*.

**Figure supplement 1.** Optic fiber placements and behavioral assessments during conditioning and retention test.

**Figure supplement 1—source data 1.** Number of bumping incidents across different groups and velocity data for *Figure 3—figure supplement 1D*.

group. Additionally, analysis of movement velocity revealed that ChR2 mice had a higher fleeing speed in response to the CS compared to the other four groups (*Figure 3D*).

Fear memory was assessed 24 hr after conditioning by presenting four CSs alone. During the retention test, the control group exhibited robust freezing as its dominant defensive behavior. In contrast, the ChR2 group displayed significantly higher fight scores compared to the other three groups (*Figure 3E and F*; *Video 2*). The Jaws group exhibited lower flight responses due to inhibited US signaling during conditioning. However, contrary to our expectations, the Jaws group displayed freezing levels that were not significantly different from those of the control group (*Figure 3E*). This result may be attributed to post-illumination rebound excitation, as Jaws has been shown to yield residual activity even when ramped illumination is used to minimize this effect (*Chuong et al., 2014*). The unpaired group exhibited significantly lower freezing levels compared to the paired control group. However, flight scores in the unpaired group were significantly higher than those in the control group. This was likely due to the tendency of the control group to remain frozen before and during the CS presentation.

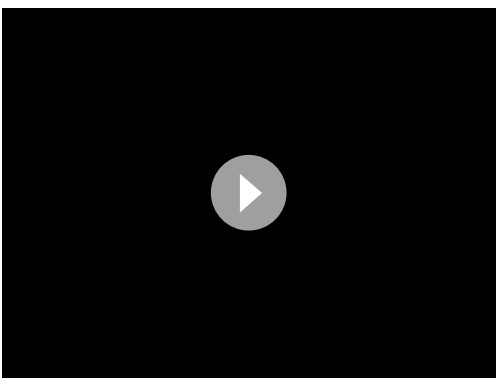

**Video 1.** Fleeing behavior during conditioning in the ChR2 group in response to conditioned stimulus (CS). The video shows the behavior of a representative animal from the control, ChR2, and Jaws groups during the last trial of conditioning. The sequence includes 10 s pre-CS, 10 s of CS (7 s of CS alone followed by 3 s of robot chasing), and 10 s post-CS. The ChR2 animal demonstrated high levels of fleeing behavior in response to the CS, compared to the control and Jaws groups.

https://elifesciences.org/articles/101523/figures#video1

The unpaired group, however, showed sensory orientation responses to the CS, contributing to their elevated fleeing scores. Moreover, the movement speed of the unpaired group during the tone CS did not exceed 3 cm/s, suggesting exploratory rather than defensive behavior in the test environment (*Figure 3—figure supplement 1D*). Taken together, these findings suggest that enhanced CGRP activity during imminent threat promotes fleeing behavior during conditioning, sustaining the heightened flight response through the retention test.

We next examined whether the same modulation of CGRP neuron activity paired with electric footshock (1 s; 0.3 mA) would also engage in active defensive behavior. After a 1 wk rest period following the conditioning paradigm with the robot, the control, ChR2, and Jaws groups of mice underwent conventional fear conditioning (*Figure 4A*). Although the CS used in the conventional fear conditioning (12 kHz; 70 dB; 10 s) differed from the one used with the robot (4 kHz; 70 dB; 10 s), residual effects were observed in the ChR2 group during the habituation session and the first block of conditioning (*Figure 4C*).

During conditioning, both ChR2 and control groups exhibited a gradual increase in freezing as the trials progressed (*Figure 4B*). However, consistent with the previous experiment with the robot, the Jaws group showed significantly lower levels of freezing compared to the other two groups. Moreover, although some high flight scores were observed in the ChR2 group during the first block of trials, all three groups exhibited equivalently low levels of fleeing responses as trials progressed (*Figure 4C*). When fear memory was tested 24 hr later, ChR2-expressing mice displayed significantly more freezing compared to both control and Jaws-expressing mice (*Figure 4D*). The Jaws and control groups exhibited similar levels of freezing, with no significant difference between the two groups. In terms of fleeing response, since all three groups demonstrated minimal fleeing responses, there was no significant difference observed (*Figure 4E*). These data show that the same CGRP stimulation did not promote fleeing responses; however, with footshock as in the US, the freezing response observed during conditioning was intensified in the retention test. Overall, additional activation of CGRP neurons enhances fear learning and memory, resulting in conditioned fleeing responses following robot chasing and conditioned freezing responses after footshock.

## CGRP neurons intensify threat perceptions and regulate defensive behaviors

To further investigate whether the previously observed fleeing responses with additional CGRP

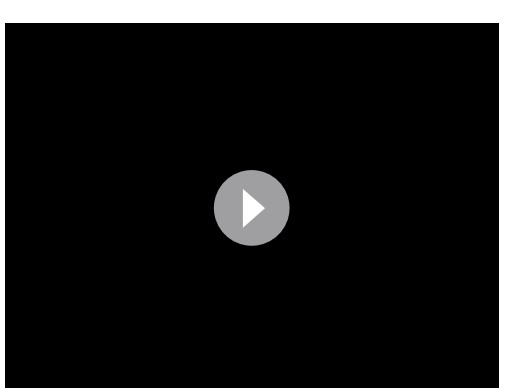

**Video 2.** Fleeing behavior during aversive memory recall in the ChR2 group in response to conditioned stimulus (CS). To test the fear memory in the retention test, we used a rectangular box instead of the donut maze used during conditioning. The box was placed on top of the donut maze, preventing the animals from seeing the robot's location. The CS was delivered from the robot's speaker, with the flashing light indicating the onset of the CS. The video shows the animals' behavior during the first trial of the retention test, where the ChR2 mouse showed more fleeing behavior compared to the other two mice.

https://elifesciences.org/articles/101523/figures#video2

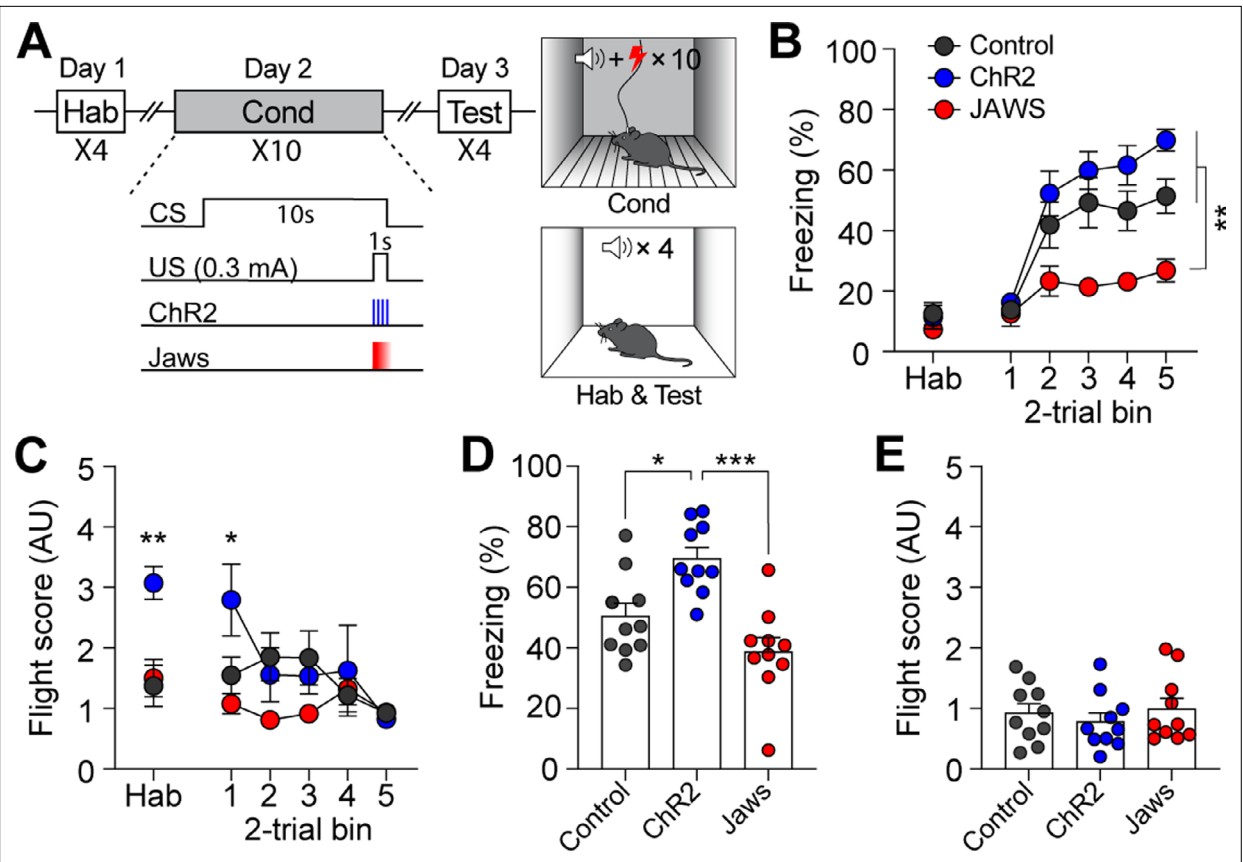

**Figure 4.** Calcitonin gene-related peptide (CGRP) activation during conventional fear conditioning promotes passive, but not active, defensive behavior. (**A**) A schematic diagram of fear conditioning protocol with the footshock. (**B**) Freezing to the conditioned stimulus (CS) during habituation and conditioning sessions. All three groups showed a progressive increase in freezing as trials progressed, but the Jaws group froze significantly less compared to the other two groups (significant group effect in a repeated-measures two-way ANOVA, F(2, 27)=19.74, p<0.001; subsequent post-hoc tests, <i>p-values <0.01). (**C**) Flight scores in response to the CS during habituation and conditioning sessions. The ChR2 group showed significantly higher fleeing responses during habituation, suggesting some residual effect from fear conditioning with the robot (one-way ANOVA, F(2, 27)=9.37, p<0.01; post-hoc test, <i>p-value <0.01). A repeated-measures two-way ANOVA revealed significant group differences during conditioning (F(2, 27)=8.91, p<0.01); however, post-hoc analysis showed that these differences were only significant in the first block of trials, where the ChR2 group exhibited higher fleeing responses than the other two groups, with no group differences observed in subsequent blocks. (**D**) Average freezing in response to the CS during the retention test. The ChR2 group froze significantly more than the Jaws and control groups (one-way ANOVA, F(2, 27)=13.31, p<0.001; post-hoc tests, <i>p-values <0.05). (**E**) Average flight scores to the CS during the retention test. Fleeing responses were minimal across all three groups, and no significant differences were observed (one-way ANOVA, F(2, 27)=0.12, p=0.63). *p<0.05, **p<0.01, ***p<0.001.

The online version of this article includes the following source data for figure 4:

**Source data 1.** Behavior data for *Figure 4*.

activation in the presence of the robot (*Figure 3C*) were due to the intensified perception of the US threat, we systematically escalated the threat level of the US by increasing the robot's speed without manipulating CGRP activity. The robot speed, ranging from 70 to 90 cm/s, was carefully selected after testing various speeds to ensure that it effectively induced a conditioned response without posing harm to the animals. In the previous experiments, the robot moved at a speed of 70 cm/s, making one-and-a-half turns in the donut maze within 3 s. By increasing the speed to 80 cm/s, the robot made two full turns, and at 90 cm/s, it made two and a half turns within the same time frame. Additionally, we analyzed the correlation between robot speed and the number of physical bumps, revealing a significant positive relationship in which higher robot speeds led to more bumps (*Figure 5—figure supplement 1A* and B). These findings suggest that the increased robot speed resulted in the animals perceiving a greater threat due to more physical contact.

On the conditioning day, all three groups showed equivalent levels of freezing behavior (*Figure 5A*). However, animals exposed to the highest speed exhibited significantly higher flight scores compared

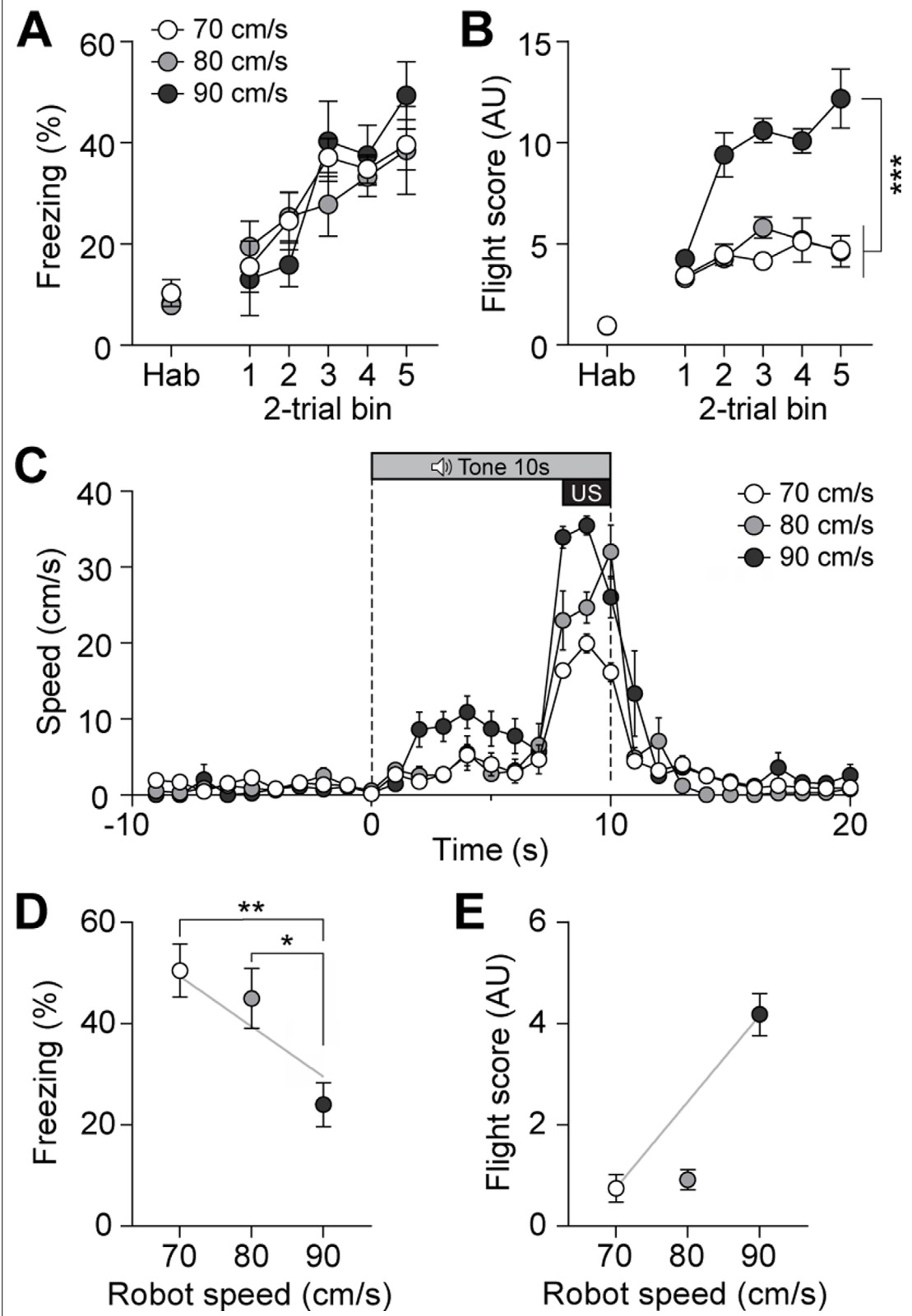

**Figure 5.** Escalating threat intensity modulates defensive behavior. (**A**) Freezing to the conditioned stimulus (CS) during habituation and conditioning sessions for groups subjected to three different robot speeds (n=8 per group). All three groups showed an equivalent progressive increase in freezing as trials progressed, with no significant differences between the groups (repeated-measures two-way ANOVA, F(2, 21)=0.11, p=0.89). (**B**) Flight scores in response to the CS during habituation and conditioning sessions. Animals exposed to 90 cm/s robot speed exhibited higher flight scores compared

*Figure 5 continued on next page*

*Figure 5 continued*

to the other two groups (repeated-measures two-way ANOVA, F(2, 21)=76.43, p<0.001; post-hoc tests, *p*-values <0.001). (**C**) Average velocities in response to the CS during the conditioning. The average velocity of the 90 cm/s group during the CS was significantly higher than that observed in the 70- and 80 cm/s groups (significant group effect in a repeated-measures two-way ANOVA, F(2, 21)=8.42, p<0.001; post-hoc test, *p*-values <0.01). (**D**) Average freezing in response to the CS during the retention test. Animals exposed to 70- or 80 cm/s robot speed froze significantly more compared to those subjected to 90 cm/s (one-way ANOVA, F(2, 21)=6.60, p<0.01; post-hoc tests, *p*-values <0.05). There was a negative correlation between freezing responses and robot speed (gray line; r=–0.61, p<0.01). (**E**) Average flight scores in response to the CS during the retention test. Animals subjected to 90 cm/s robot speed displayed significantly higher flight scores compared to those exposed to 70- and 80 cm/s (one-way ANOVA, F(2, 21)=58.09, p<0.001; post-hoc tests, *p*-values <0.001). Fleeing responses were positively correlated with robot speed (gray line; r=0.82, p<0.001).

The online version of this article includes the following source data and figure supplement(s) for figure 5:

**Source data 1.** Behavior data for *Figure 5*.

**Figure supplement 1.** Effects of robot speed on bumping incidents and behavioral responses.

**Figure supplement 1—source data 1.** Number of bumping incidents across groups at different robot speeds and velocity data for *Figure 5—figure supplement 1C*.

to the other two groups (*Figure 5B*). Analysis of movement further revealed that animals in the 90 cm/s condition exhibited the highest speeds in response to the CS compared to those in the 70 cm/s and 80 cm/s groups (*Figure 5C*). On the retention test day, no group differences in freezing were observed between the 70 cm/s and 80 cm/s groups, but 90 cm/s group exhibited significantly lower freezing levels than the other two groups (*Figure 5D*). In contrast, animals in the 90 cm/s speed displayed a significantly higher number of flight scores compared to the other two groups (*Figure 5E*). There was a positive correlation between robot speed and fleeing responses, and a negative correlation between robot speed and freezing responses (*Figure 5D and E*). Furthermore, animals exposed to the 90 cm/s speed exhibited a fleeing response similar to that of those subjected to 70 cm/s robot chasing with CGRP stimulation (*Figure 3C and F*). Considering the combined results of these experiments, the increased activity of CGRP neurons likely enhances fleeing behavior by amplifying the perceived threat of the US.

We next sought to confirm whether CGRP neurons are necessary for inducing active defensive behavior under high-speed conditions. Since Jaws inhibition was insufficient to block fear memory formation (*Figures 3E and 4D*), we bilaterally injected either Cre-dependent tetanus toxin light chain (TetTox; AAV-DIO-GFP:TetTox) for effective silencing by selectively blocking neurotransmitter release, or AAV-DIO-eYFP (control) into the PBN (*Figure 6A*; *Jo et al., 2020*). Mice then underwent fear conditioning with the robot at a speed of 90 cm/s. On the conditioning day, the TetTox group exhibited significantly lower levels of both freezing and fleeing compared to the control group (*Figure 6B and C*). Moreover, velocity analysis confirmed that the TetTox group, with silenced CGRP neurons, showed little to no fleeing behavior in response to the CS, while the control group exhibited robust flight responses (*Figure 6D*). This persisted on the retention day, with the TetTox group consistently showing reduced levels of freezing and fleeing compared to controls (*Figure 6E and F*). These results suggest that CGRP neurons are necessary for perceiving threats and promoting fleeing. Enhancing CGRP neuronal activity, either optogenetically or by increasing threat levels, strengthens fear learning and memory, leading to intensified active defensive behaviors.

## Discussion

CGRP neurons, known for relaying US information to the forebrain and inducing passive freezing behavior (*Han et al., 2015*), were examined to explore their role in active defensive responses. Using a naturalistic paradigm with a robot and other aversive stimuli of varying threat levels, we recorded neuronal activity and found that CGRP neurons encode different threat intensities through variations in firing duration and amplitude. Optogenetic activation of these neurons during fear conditioning amplified the perceived threat posed by the US, making it seem more dangerous, while inhibiting them weakened it. The expression of defensive behaviors varied depending on the type of US: under robot chasing conditions, increased CGRP activity made the robot seem as if it were moving faster, driving the animals to flee more during both conditioning and the retention test. In contrast, under footshock conditions, elevated CGRP activity predominantly enhanced freezing responses. Systematically escalating robot speed to simulate higher threat levels strengthened conditioned fleeing

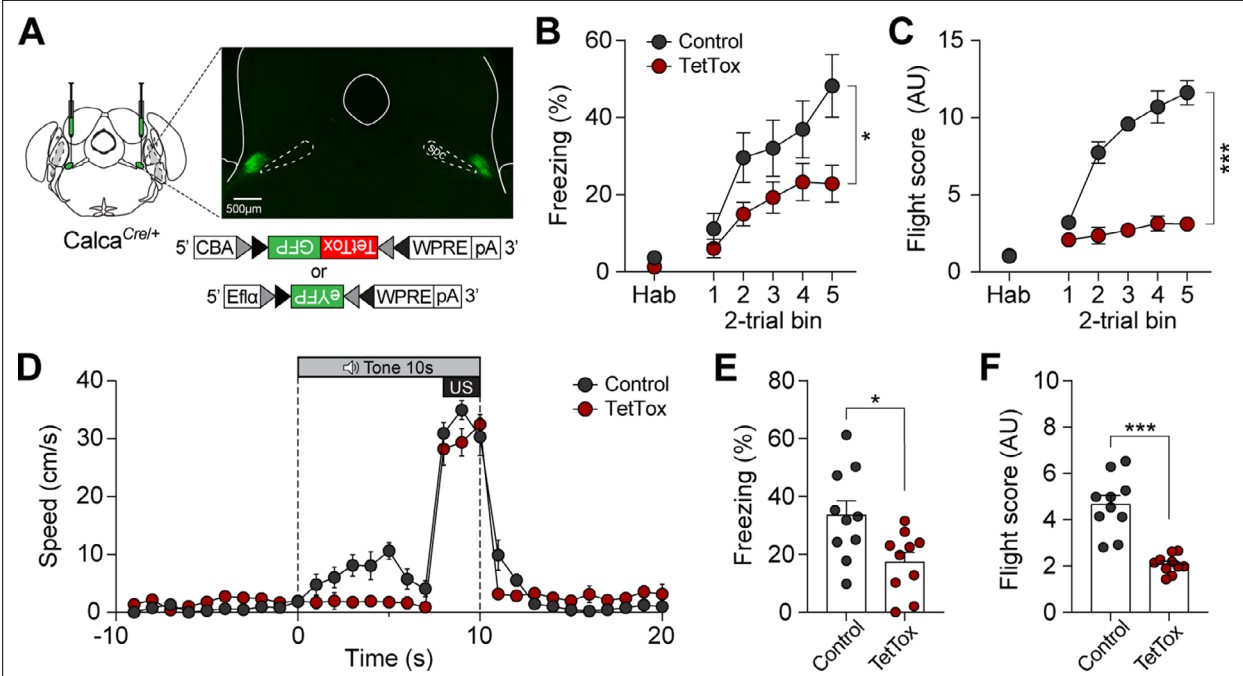

**Figure 6.** Calcitonin gene-related peptide (CGRP) neurons are necessary for promoting active defensive behaviors under high-speed threat conditions. (**A**) Schematic of bilateral AAV-DIO-GFP:TetTox or AAV-DIO-eYFP (control) injections into the parabrachial nucleus (PBN) and representative histological images of TetTox expression. (**B**) Freezing to the conditioned stimulus (CS) during habituation and conditioning sessions (n=10 per group). Both groups showed a progressive increase in freezing, but the TetTox group, with inactivated CGRP neurons, exhibited significantly lower levels of freezing compared to the control group (significant group effect in a repeated-measures two-way ANOVA, F(1, 18)=6.42, p<0.05). (**C**) Flight scores in response to the CS during habituation and conditioning sessions. The control group showed significantly greater levels of fleeing responses compared to the TetTox group (significant group effect in a repeated-measures two-way ANOVA, F(1, 18)=235.27, p<0.001). (**D**) Average velocities in response to the CS during the conditioning. Control animals displayed significantly higher flight scores compared to the TetTox group (repeated-measures two-way ANOVA, F(1, 18)=31.91, p<0.001). (**E**) Average freezing to the CS during the retention test. The TetTox group displayed significantly lower levels of freezing compared to the control group (independent t-test, t(18) = 2.7, p<0.05). (**F**) Average flight scores in response to the CS during the retention test. The TetTox group showed significantly lower levels of fleeing compared to the control group (independent t-test, t(18) = 6.29, p<0.001). *p<0.05, ***p<0.001.

The online version of this article includes the following source data and figure supplement(s) for figure 6:

**Source data 1.** Behavior data for *Figure 6*.

**Figure supplement 1.** Effects of CGRP neuron silencing on bumping and CS-evoked velocity.

**Figure supplement 1—source data 1.** Number of bumping incidents and velocity data for *Figure 6—figure supplement 1B*.

responses, whereas silencing these neurons prevented the formation of active defensive behaviors. Overall, our findings indicate that CGRP neurons act as a general alarm system, primarily orchestrating freezing responses but also contributing to active defensive behaviors by intensifying the perception of heightened threats, ensuring responses are adapted to the level of danger.

The choice of defensive strategies depends on the perceived severity of the threat (*Fanselow and Lester, 2013*). For instance, when an animal detects a predator at a relatively safe distance, freezing is the most likely defensive behavior, as it helps avoid detection. However, as the threat becomes more imminent and threat levels increase, freezing is no longer the optimal choice. At this point, the animal shifts from passive freezing to more active defense, adopting behaviors such as fleeing or, if necessary, fighting. However, most research on CGRP has utilized footshock (*Bowen et al., 2020*; *Han et al., 2015*) or other aversive stimuli in small arenas (*Kang et al., 2022*), potentially limiting the observation of fleeing responses and leading to a focus on passive freezing as the predominant behavior studied. In our study, we introduced different types of US designed to present an imminent threat. This approach allowed animals to perceive the threat through dynamic sensory inputs, with the distance to the threat being discernible. By incorporating the predator-like robot, we developed a behavioral paradigm that enabled animals to form fear memory while minimizing differences in bumping incidents between groups. Additionally, this paradigm allowed for the observation of two

distinct defensive responses to the CS: passive freezing and active fleeing, providing a platform to explore defensive behaviors across varying threat contexts.

Recent studies have modified conventional fear conditioning protocols to investigate active defensive behaviors in animals (*Borkar et al., 2024*; *Fadok et al., 2017*). One such example is changing the CS to a serial-compound stimulus, where a pure tone is immediately followed by a white noise, inducing freezing and flight responses, respectively. While effective for observing transitions between freezing and fleeing, our electrophysiological data show that CGRP neurons are more excited in response to the US compared to the CS (*Figure 1F and G*). Thus, altering the type of US is more appropriate for studying CGRP neurons. In addition, the robot allowed us to systematically increase threat levels by adjusting its speed, providing a more controlled approach to studying defensive behaviors. Our results showed that a robot speed of 70 cm/s did not induce fleeing response during the retention test (*Figures 3F and 5E*); however, increasing the robot's speed to 90 cm/s elicited conditioned flight responses in control mice. Moreover, CGRP activation combined with a 70 cm/s robot speed induced flight responses similar to those observed with a 90 cm/s robot speed in control mice. This suggests that CGRP activation amplifies the perceived threat, thereby promoting active defensive behaviors.

We optogenetically inhibited CGRP neurons in animals while they were being chased by the robot during conditioning. During this session, the Jaws group showed less freezing and fleeing compared to the control group. However, the reduced fear responses were not sustained on the retention test day, suggesting that transient inhibition during the chasing was insufficient to suppress the acquisition of fear memory (*Figure 3E*). It has been reported that a square pulse can cause strong rebound excitation following inhibition (*Chuong et al., 2014*). While ramped illumination reduces the magnitude of rebound excitation, it does not eliminate it entirely, leaving small residual excitation. This residual activity may have contributed to the formation of fear memory despite the inhibition of CGRP neurons. In a subsequent experiment with increased robot speed, we used TetTox to silence CGRP neurons more effectively compared to temporary inhibition. This group consistently showed lower fear responses compared to the control group even on the retention day, indicating a more significant impact on fear learning compared to transient inhibition. However, the progressive increase in freezing levels across trials during conditioning, even with CGRP neurons silenced (*Figure 6B*), suggests the involvement of other pathways in processing the aversive stimuli. For instance, different populations of CGRP neurons in the parvocellular subparafascicular nucleus of the thalamus also respond to threats and relay negative emotional signals to the amygdala thereby contributing to aversive memory formation (*Kang et al., 2022*). Additionally, the midbrain periaqueductal gray (PAG) transmits aversive signals to the amygdala (*Johansen et al., 2010*; *Johansen et al., 2012*; *Ozawa et al., 2017*) and other forebrain structures (*Lefler et al., 2020*; *Esteban Masferrer et al., 2020*), ensuring effective expression of defensive responses upon the detection of a threat. Although fear learning can occur through various pathways, our use of optogenetics and TetTox suggested that CGRP neurons contribute to both active and passive defensive behaviors, facilitating responses that are appropriate to the magnitude of the threat.

In conclusion, by employing both conventional footshock and a naturalistic paradigm, the present study emphasizes the role of CGRP neurons in facilitating both passive and active defensive behaviors. Optogenetic stimulation of CGRP neurons in the absence of external stimuli induced robust freezing, and their activation during conventional fear conditioning further amplified conditioned freezing, demonstrating their primary role in driving passive defensive responses. However, under heightened threat conditions, such as enhanced CGRP activation or faster robot speeds, these neurons also strengthened active defensive behaviors by amplifying perceived threat. These findings suggest that CGRP neurons detect and process threats, predominantly driving freezing behavior, while also enabling active responses under heightened danger to facilitate appropriate defensive behaviors aligned with the intensity of the threat.

# Materials and methods

## Key resources table

| Reagent type (species) or resource | Designation | Source or reference | Identifiers | Additional information |
|---|---|---|---|---|
| Strain, strain background (*Mus musculus*) | B6.Cg-Calca[tm1.1(cre/EGFP)Rpa]/J | The Jackson Laboratory | RRID:IMSR_JAX:033168 | |
| Strain, strain background (C57BL/6 J, C57BL/6 J) | C57BL/6 J | The Jackson Laboratory | RRID:IMSR_JAX:000664 | |
| Recombinant DNA reagent | AAV-DIO-ChR2-mScarlet | IBS Virus Facility | N/A | |
| Recombinant DNA reagent | AAV-DIO-Jaws-GFP | *Jo et al., 2018* | N/A | |
| Recombinant DNA reagent | AAV-DIO-eYFP | Addgene | RRID:Addgene_27056 | |
| Recombinant DNA reagent | AAV-DIO-TetTox-GFP | *Han et al., 2015* | N/A | |
| Software, algorithm | ANY-maze 5.3 | Stoelting Co | N/A | |
| Software, algorithm | GraphPad Prism | GraphPad software | N/A | |
| Software, algorithm | SPSS | IBM | N/A | |
| Software, algorithm | Offline Sorter | Plexon Inc | N/A | |

## Animals

We used heterozygous *Calca*[Cre/+] mice, generated by breeding *Calca*[Cre/Cre] (Cat. 033168) with C57BL/6 J (Cat. 000664) from Jackson Laboratory. Both male and female mice, aged 3–6 mo, were used in all studies, and no sex differences were observed. Mice were housed in a temperature- and humidity-controlled facility on a 12 hr light/dark cycle (lights off at 7 AM) with ad libitum access to food and water. All experiments were performed during the dark phase of the cycle under the guidelines of the Institutional Animal Care and Use Committee at the Korea University (KUIACUC-2022–0057).

## Virus production

All AAV vectors were prepared as described previously (*Pyeon et al., 2024*). Cre-dependent optogenetic viruses included AAV-DIO-ChR2-mScarlet, AAV-DIO-Jaws-GFP, and AAV-DIO-eYFP (control and unpaired). For selective inactivation of CGRP neurons, AAV-DIO-TetTox-GFP was used. Viral aliquots were stored at –80°C before stereotaxic injection.

## Stereotaxic surgery

Mice were anesthetized with isoflurane (4% induction, 1.5–2% maintenance) and fixed on a stereotaxic frame (Model 942, David Kopf Instruments). After exposing the skull, bregma and lambda were aligned on the same horizontal plane. Small burr holes were then made for viral injections and optic fibers, and additional holes were drilled for anchoring screws. Cre-dependent virus (0.5 µl per side) was injected unilaterally or bilaterally into the PBN (5.0 mm posterior, 1.5 mm lateral, and 3.5 mm ventral to bregma) at a rate of 0.25 µl /min. Microdrives or optic fibers (200 µm diameter, 0.22 numerical aperture) were implanted 0.3 mm dorsal to the virus injection sites and secured with dental cement. Meloxicam (1.5 mg/kg) was administered subcutaneously to alleviate pain and reduce inflammation. Mice were allowed to recover for 2–3 wk before the start of behavioral experiments.

## 30s stimulation of CGRP stimulation

Two weeks after surgery, the optic fibers implanted in the mice were connected to optic cables, and the animals were placed in an open arena (30 × 22 × 22 cm). After 2 min of exploration, CGRP neurons were either stimulated or inhibited four times at 60 s intervals. For activation, 40 Hz of blue light (473 nm; LaserGlow) was delivered for 30 s. For inactivation, continuous red light (640 nm; LaserGlow) was delivered for 3 s followed by a 1 s ramp down, repeated in cycles until a total duration of 30 s. The light output from the bilateral branching cable was set to 9±0.5 mW. The animals' behavior was recorded using a camera mounted on the ceiling of the chamber. Freezing and movement velocities were analyzed using video-tracking software (ANY-maze, Stoelting Co.).

## Fear conditioning with chasing robot

Fear conditioning experiment was conducted using a box-shaped robot (15 × 26 × 35 cm) with four high-traction wheels that moved quickly inside a white acrylic track (18 cm width) of a donut-shaped maze (60 cm outer diameter). The speed of the robot was controlled by a Bluetooth-based microcontroller with a custom-written program in Arduino, and the CS (4 kHz; 80 dB; 10 s) was generated by speakers mounted in the front and back of the robot. Habituation and fear responses to the CS were tested before and after conditioning in a rectangular box (30 × 27 × 20 cm). The maze and rectangular box were wiped with 70% ethanol between animals. The custom-written program required for the robot's operation is provided in the source code file 1.

On the first day of fear conditioning paradigm, habituation to the CS was performed. Mice were introduced to the rectangular box and allowed a 3 min exploratory period, followed by the presentation of 4 CSs at intervals of 60 s. During this phase, the chasing robot was positioned outside the rectangular box to prevent the animals from seeing it. On day 2, optic fiber cables were attached to the head of each mouse, which were then placed in the donut-shaped maze. After 3 min, mice received 10 associations of a 10 s CS, each co-terminating with 3 s of chasing (speed of 70, 80, or 90 cm/s) with 60 s interval. The robot chased animals at high speeds and posed a physical threat by colliding and pushing them. During robot chasing, CGRP neurons were either activated or inhibited. To optogenetically stimulate, 30 Hz of blue light was delivered for 3 s with the robot. For inhibition, continuous red light was delivered for 3 s followed by a 1 s ramp down with the robot. For the unpaired group, the same number of CSs was presented; however, the robot chased the animals at a random time within the ITI. On day 3, fear response to the CS was measured. Mice were placed in the same rectangular box used on the first day, and after 3 min, the CS was presented alone 4 times at 60 s intervals. During conditioning procedures, the animals' behavior was recorded using a camera mounted on the ceiling of the chamber. Freezing and movement velocities were analyzed using ANY-maze. Freezing behavior was automatically detected when movement was absent for at least 0.8 s. For fleeing responses, flight score was calculated following previous studies (*Borkar and Fadok, 2021*; *Fadok et al., 2017*). Speed (cm/s) was extracted using the animal's center body point, and the flight score was measured by dividing the average speed during each CS alone period by the speed during the same length of the pre-CS period. For conditioning trials, the CS alone period was the first 7 s from CS onset, while for the habituation and retention tests, it was 10 s. Vertical movements, such as jumping, were manually recorded by an experimenter blind to the group assignments, with 1 point added to the flight score for each escape jump. The number of times the mice bumped into the robot was manually scored by an experimenter who was blinded to the group assignments of the animals. We divided each group into two to three animal batches and replicated the experiments to confirm the consistency of the results across these batches.

## Fear conditioning with electrical footshock

A standard fear conditioning paradigm with electric footshock was conducted in four identical chambers (21.6 × 17.8 × 12.7 cm; Med Associates) placed inside sound-attenuating boxes. Each chamber was equipped with two speakers on one wall with 24 shock grids on the floor wired to a scrambled shock generator. Tone habituation and retention test of fear memory was tested in a different context where white plastic panels (20 × 16 × 12 cm) were inserted inside the chamber covering the walls and grids. The chamber and inserts were cleaned with 70% ethanol between animals.

After a week of resting period from fear conditioning paradigm with the robot, animals underwent conventional fear conditioning paradigm. On day 1, mice were habituated to a different CS (12 kHz; 80 dB; 10 s). The white plastic panels were inserted inside the chamber, and the animals were allowed to freely explore the context for 3 min. The CS was then presented four times with an ITI of 60 s. On the next day, after 3 min of free exploration, the animals received 10 CS–US trials, each co-terminating with a 1 s footshock (0.3 mA) with a 60 s ITI. CGRP neurons were either activated or inhibited during footshock delivery. For activation, 30 Hz of blue light was delivered during footshock presentation. For inhibition, continuous red light was delivered for 1 s followed by a 1 s ramp down during footshock presentation. To add a context-specific odor, a petri dish filled with a 1% acetic acid solution was placed under the grid floor. On day 3, fear memory in response to the CS was tested. As on the first day, animals were placed in the chamber with the white plastic panels, and the CS was presented four times at 60 s intervals. Animal behavior was recorded during the

experiments by a camera installed on the ceiling, and freezing and fleeing responses were analyzed afterward.

### Single-unit recording

A custom-made microdrive containing four tetrodes (20 μm diameter tungsten wire; California Fine Wire) glued to one optic fiber (200 μm core diameter, 0.22 numerical aperture) was used. Tetrode tips were cut to protrude beyond the optic fiber by 400–500 μm and were gold-plated to reach impedances of 200–500 kΩ, tested at 1 kHz. After the recovery period from surgery, individual mice were placed in a holding cage, and single-unit activity was monitored using a Cheetah data acquisition system (Digital Lynx SX, Neuralynx). Neural signals were filtered between 0.6 and 6 kHz, digitized at 32 kHz, and amplified 1000–8000 times. To identify ChR2-expressing units in the PBN, 10 blue light pulses (473 nm; 5 ms width; 4–10 mW/mm$^2$ intensity; Laswerglow technologies) were delivered at 30 Hz via the optic fiber. If no light-responsive units were detected, the tetrodes were lowered by 40–80 μm increments, up to 160 μm per day. Once light-responsive units were found, behavioral recording sessions began.

During daily recording sessions, spontaneous spikes from PBN neurons were recorded in the home cage for 10 min. Neuronal firing rates were further recorded during fear conditioning sessions over three consecutive days. On day 1, mice were first habituated to a tone (10 kHz, 80 dB, 10 s duration) as the conditioned stimulus (CS) for 10 times. On day 2, mice underwent 10 exposures to the CS, each co-terminated with an unconditioned aversive stimulus (US) consisting of 3 s of chasing by a robot, with an average inter-trial interval (ITI) of 100 s. On day 3, mice were tested for fear retention with 10 presentations of the CS alone. At the end of each recording session, 10 trains of 10 light pulses (total 100 presentations; 30 s intervals) were delivered to identify ChR2-expressing CGRP neurons in the PBN. The tetrodes were kept in the same location to compare neuronal responses to the CS across 3 d of conditioning. However, neurons recorded across 3 d were considered independent units rather than the same units.

After completing the fear conditioning sessions, neuronal firing rates were recorded in response to three different aversive stimuli: pinprick, tail pinch, and robot chasing. Pinprick and tail pinch were administered by a trained experimenter throughout all recording sessions to ensure minimal variability, as described previously (*Pyeon et al., 2024*). For the pinprick, mice were placed in a white cylindrical Plexiglass container (14 cm in diameter, 20 cm in height) with a plastic grid floor, and hind paws were pinpricked with a 26 G syringe needle (approximately 0.5 s duration). For the tail pinch, mice were placed in a rectangular Plexiglass cage (27 × 18 × 8 cm), and the tail was pinched using forceps (1 s duration). For the robot chasing, mice were chased by the same robot used in the conditioning sessions but without the predictive CS. After the daily recording session, all tetrodes were lowered by 40–80 μm to find different light-responsive neurons and the mouse was returned to its home cage.

Neuronal spikes were isolated based on various waveform characteristics using Offline Sorter (Plexon). Stably firing units throughout the behavioral recording session were further analyzed using MATLAB software (MathWorks). To classify CGRP neurons, peri-event time histograms (PETHs; 11.11 ms bins) were constructed around the light presentations. Spike probability and latency were calculated for individual units in response to a total of 100 light pulses, and a cluster analysis was conducted on all units. The cluster with the highest spike probability (>0.8) and the shortest latency (<5.5 ms) was identified as CGRP neurons. These neurons also showed higher correlations between spontaneous and light-evoked waveforms, compared with optically insensitive PBN neurons. To further examine responses of CGRP neurons to aversive stimuli, PETHs (50 s-ms bins) were generated around the time of these aversive stimuli. Firing rates in PETHs were converted to z-scores relative to baseline firing rates observed during 3 s period before each stimulus. Average neuronal responses to aversive stimuli were measured during 0.65 s window from stimulus onset.

### Histology

After completion of all behavioral experiments, mice were anesthetized and transcardially perfused with phosphate-buffered saline (PBS) followed by 4% paraformaldehyde (PFA). Dissected brains were post-fixed overnight in 4% PFA, then cryoprotected in 30% sucrose in PBS at 4 °C for 72 hr. Brains were frozen and sectioned into 30 μm coronal slices on a cryostat (CM1860, Leica Biosystems).

Sections were mounted on microscopic slides and cover-slipped with DAPI Fluoromount-G (Southern Biotech). Using a fluorescence microscope (EVOS M5000, Thermo Fisher Scientific), images were taken to examine recording sites, fiber placements, and fluorescent expression levels.

### Statistical analysis

Statistical analyses were performed using a statistical software package (SPSS version 27.0, IBM SPSS, Armonk, NY). Statistical tests for electrophysiological and behavioral results were assessed with mixed-design ANOVA that contained within-subjects variables (e.g. trials) and between-subjects variables (e.g. group) as well as one-way ANOVA across groups. Once significant interactions were observed, Bonferroni corrections were used for post hoc pairwise comparisons. Two-tailed p-values <0.05 were considered significant. All data were expressed as mean ± SEM. All statistical results are summarized in the *Supplementary file 1*.

## Acknowledgements

This work was supported by the National Research Foundation of Korea grant funded by the Korean government (2022M3E5E8017804 to YSJ).

## Additional information

### Funding

| Funder | Grant reference number | Author |
| --- | --- | --- |
| National Research Foundation of Korea | 2022M3E5E8017804 | Yong Sang Jo |

The funders had no role in study design, data collection and interpretation, or the decision to submit the work for publication.

### Author contributions

Gyeong Hee Pyeon, Conceptualization, Software, Formal analysis, Investigation, Writing - original draft, Writing - review and editing; Hyewon Cho, Software, Investigation; Byung Min Chung, Formal analysis, Investigation; June-Seek Choi, Writing - original draft, Project administration; Yong Sang Jo, Conceptualization, Formal analysis, Supervision, Funding acquisition, Investigation, Writing - original draft, Project administration, Writing - review and editing

### Author ORCIDs

Gyeong Hee Pyeon https://orcid.org/0000-0003-4013-3206
June-Seek Choi https://orcid.org/0000-0002-4394-2140
Yong Sang Jo https://orcid.org/0000-0002-7716-0964

### Ethics

All experiments were performed during the dark phase of the cycle under the guidelines of the Institutional Animal Care and Use Committee at the Korea University (KUIACUC-2022-0057).

### Decision letter and Author response

Decision letter https://doi.org/10.7554/eLife.101523.sa1
Author response https://doi.org/10.7554/eLife.101523.sa2

## Additional files

### Supplementary files

MDAR checklist

Supplementary file 1. Statistical summary of behavioral and neural data.

Source code 1. Arduino code for controlling the chasing robot.

## Data availability

All data generated or analyzed during this study are included in the manuscript and supporting files.

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
