## [Editor Report]

This valuable work advances our understanding of parabrachial CGRP threat function. The evidence supporting CGRP aversive outcome signaling to promote active defensive behavior is solid. The work will be of interest to neuroscientists studying defensive behaviors.

---

## [Decision Letter]

**Decision letter after peer review:**

Thank you for submitting your article "Parabrachial CGRP neurons modulates conditioned active defensive behavior under a naturalistic threat" for consideration by *eLife*. Your article has been reviewed by 3 peer reviewers, and the evaluation has been overseen by a Reviewing Editor and Kate Wassum as the Senior Editor.

Essential revisions (for the authors):

Three reviewers have assessed your manuscript and are asking that you revise and resubmit your findings. While all three reviewers saw the experiments as addressing an important topic, there were a variety of concerns about the experimental design and statistical reporting. These concerns must be fully addressed in any resubmission.

The biggest concern was how fleeing is being reported – by flight count. Flight counts are very low and may not be amendable to standard statistical analyses. Reporting an additional flight metric, such as speed, may provide a better overall picture of the behavior and be more amendable to statistical analyses.

Another concern was the design and reporting of behavior during the predator-like robot experiments. First, these experiments did not use any sort of control cue, so it is unclear to what degree cue-evoked behavior is due to the relationship with predator rotation. Second, from the manuscript it is totally unclear how the mice behaved during this session, and how they are interacting with the predator. Are there frequent collisions? What is the pattern of fleeing like during these sessions?

Finally, experimental evidence showing that CGRP neuron activation is necessary and sufficient to produce flight is not substantiated. Instead, manipulations are focused on altering the intensity of the unconditioned stimulus – the predator-like robot.

In addition to these consensus concerns, several other concerns were raised by individual reviewers. These concerns should also be addressed in any resubmission.

*Reviewer #1 (Recommendations for the authors):*

Excellent work. My only recommendation is to use control (unpaired) procedures in the robot US setting. Need to be confident your observed flight behavior is due to pairing of the CS and US.

*Reviewer #2 (Recommendations for the authors):*

I would like to congratulate the authors on a well-executed and insightful study. The innovative approach and comprehensive analysis presented in this manuscript make a significant contribution to our understanding of the role of CGRP neurons in defensive behaviors. The findings are robust and clearly demonstrate the dual role of these neurons in modulating both passive and active responses to threats. The manuscript is well-written, the methods are sound, and the results are compelling. I believe this work will have a meaningful impact on the field and offer valuable insights for future research. I commend the authors for their excellent work and have no further suggestions at this time.

*Reviewer #3 (Recommendations for the authors):*

1. In this study, Pyeon et al. employ a predator-like robot to chase the animals to simulate naturalistic threat imminence in mice. While the idea and attempt are well intentioned, the overall data suggests the model is only inducing weak unconditional responses to the robot. Mice show only a small number of fleeing attempts compared to the number of robot chase trials.

2. For the conventional threat conditioning experiment, the authors subjected previously robot-conditioned mice to sequential conditioning. This resulted in a higher fleeing count in the ChR2 group during the habituation session (Figure 4 C). This effect could be due to a sensitization effect caused by previous ChR2 stimulation of the CGRP neurons.

3. The speed of the robot used in this study (70-90 cm/s) is quite high and is likely to cause frequent collisions with the mouse. The paper does not describe what happens if the robot collides with the mouse. Does it stop? Does it push the mouse forward? The behavior during the conditioning session needs to be reported in much more detail. Also, the small number of flight bouts in all experiments suggests that the robot is not being perceived as a high-intensity threat.

4. It is exceedingly difficult to discern how fleeing counts are defined and quantified. Because fleeing is being reported as counts, this indicates that these are discrete data, therefore, ANOVA is not an appropriate statistical method. ANOVAs assume continuous data and a normal distribution.

5. In Figure 3 B-C, ChR2 stimulation did not further elevate the freezing or fleeing count during the conditioning phase. Since US-mediated neuronal activation was elevated for at least 3 sec post termination of the US (Figure 1F, H), ChR2 stimulation should be performed for such an extended period. If possible, authors should also present the post-US data (do the mice freeze after the robot?).

6. The intensity of the pinprick and tail pinches are not defined and are potentially inconsistent across the mice as these stimuli are administered manually.

7. On page 9, the authors state the following about the ChR2 stimulation, "Combined with the spontaneous firing rate during robot chasing (about 40 Hz), this totaled approximately 70 Hz, matching the peak firing rate observed in the upper quartile during tail pinch". ChR2 simulation will entrain the neuron to fire with each pulse up to approximately 20 Hz, after which the failure rate increases. In effect, neurophysiology does not work in an additive fashion.

8. The authors claim, "We hypothesized that CGRP neurons contribute to the selection of adaptive defensive behaviors by signaling the severity of the threat". The selection of defensive behavior is not tested in these experiments

[Editors' note: further revisions were suggested prior to acceptance, as described below.]

Thank you for resubmitting your work entitled "Parabrachial CGRP neurons modulate conditioned active defensive behavior under a naturalistic threat" for further consideration by *eLife*. Your revised article has been evaluated by Kate Wassum (Senior Editor) and Michael McDannald (Reviewing Editor).

The three reviewers agree that the revisions made strengthened the manuscript. However, additional revisions are needed. These are addressed in full below. The most important changes are clarifying language about the non-associative nature of flight and the role for CGRP neurons in flight vs US activity.

*Reviewer #1 (Recommendations for the authors):*

In my initial review I expressed concern that the flight observed to the cue was not due to pairing with the chasing robot, but would be observed to any cue played in the context. In response, the authors ran a cohort of unpaired mice which received cue presentations and the chasing robot at different times. I appreciate the authors including this control group. However, as was my concern, there is very poor evidence that flight in this setting is due to pairing with the chasing robot. Within the chasing sessions, paired and unpaired mice are showing comparable levels of flight. Further, in the extinction test unpaired mice appear to show even greater flight than the paired mice. By contrast, freezing shows specificity to paired mice and is readily observed during extinction testing. This is problematic for flight because it shows that the same mice not showing conditioned flight show conditioned freezing. Recent studies in both mice and rats that have used appropriate control cues have observed robust conditioned flight. That bar is not reached in this manuscript.

Because the authors addressed my concerns, I am positively inclined towards the manuscript. However, because there is very little evidence that conditioned active defensive behaviors were observed, the authors need to refrain from using the term in the title and the manuscript. Further, the authors need to describe flight in a way that does not describe or suggest that it is conditioned or learned. This only seems correct given their observations of unpaired responding.

*Reviewer #3 (Recommendations for the authors):*

The manuscript by Pyeon et al. has been improved by the addition of additional analysis, an unpaired control condition, and clarification of experimental design. I have a few remaining concerns which should be addressed.

1. There remain numerous instances in the text that imply that CGRP neurons are capable of inducing escape behavior (Lines 59-60; 71-72; 260-262; 315-316), yet this is not demonstrated experimentally. In fact, when CGRP neurons are optogenetically activated in the absence of external stimuli, they induce freezing behavior, not flight (Figure 2; Lines 142-143). What the authors show is that CGRP neurons are important for US encoding, which under the right experimental conditions can allow the CS to elicit flight behavior (i.e., under high threat conditions). The authors should change their language or demonstrate that CGRP neurons can elicit flight. Further to this point, the TetTox approach does not allow for the disambiguation of CGRP neuron involvement in different phases of learning or expression (Lines 312-313).

2. The authors describe what happens when the robot collides with the mouse (e.g., Lines 89-90). The authors should provide a video demonstrating this, and I would like to see a quantification of how the animal's speed is altered by collision. The authors also state that bumping leads to stronger conditioned defensive behaviors (Lines 155-156) but these data are not shown. This correlation should be shown.

3. The argument given in Lines 174-176 should be rephrased to indicate that this is a hypothesis for why the Jaws group had freezing similar to control levels.

4. The authors claim that collisions with the robot push the mouse forward but do not harm the animal, yet in Lines 298-299 they posit that prolonged neuronal excitation was "presumably due to the pain experienced after bumping into robot". Which is it, painful or innocuous?

5. A more accurate Figure title for Figure 3 would be "Activation of CGRP neurons during US presentation enhances active defensive behavior".

6. Figure 5 needs a new title as CGRP neurons are not manipulated or recorded in these experiments.

7. The authors should comment on the fact that the TetTox mice react normally to US exposure (Figure 6). Which pathways are mediating the US behavioral response?

---

## [Author Response]

Essential revisions (for the authors):Three reviewers have assessed your manuscript and are asking that you revise and resubmit your findings. While all three reviewers saw the experiments as addressing an important topic, there were a variety of concerns about the experimental design and statistical reporting. These concerns must be fully addressed in any resubmission.The biggest concern was how fleeing is being reported – by flight count. Flight counts are very low and may not be amendable to standard statistical analyses. Reporting an additional flight metric, such as speed, may provide a better overall picture of the behavior and be more amendable to statistical analyses.Another concern was the design and reporting of behavior during the predator-like robot experiments. First, these experiments did not use any sort of control cue, so it is unclear to what degree cue-evoked behavior is due to the relationship with predator rotation. Second, from the manuscript it is totally unclear how the mice behaved during this session, and how they are interacting with the predator. Are there frequent collisions? What is the pattern of fleeing like during these sessions?Finally, experimental evidence showing that CGRP neuron activation is necessary and sufficient to produce flight is not substantiated. Instead, manipulations are focused on altering the intensity of the unconditioned stimulus – the predator-like robot.In addition to these consensus concerns, several other concerns were raised by individual reviewers. These concerns should also be addressed in any resubmission.Reviewer #1 (Recommendations for the authors):Excellent work. My only recommendation is to use control (unpaired) procedures in the robot US setting. Need to be confident your observed flight behavior is due to pairing of the CS and US.

We appreciate the reviewer's positive feedback and valuable recommendation. In response, we have included an unpaired control group in the robot US setting, and the updated data are now presented in Figure 3. We believe that this addition further strengthens our conclusion that the observed flight behavior is a result of associative learning between the CS and US, rather than a general reaction to the experimental conditions.

Reviewer #2 (Recommendations for the authors):I would like to congratulate the authors on a well-executed and insightful study. The innovative approach and comprehensive analysis presented in this manuscript make a significant contribution to our understanding of the role of CGRP neurons in defensive behaviors. The findings are robust and clearly demonstrate the dual role of these neurons in modulating both passive and active responses to threats. The manuscript is well-written, the methods are sound, and the results are compelling. I believe this work will have a meaningful impact on the field and offer valuable insights for future research. I commend the authors for their excellent work and have no further suggestions at this time.

We sincerely thank the reviewer for the thoughtful and positive evaluation of our work. We appreciate the recognition of the significance of our findings, which encourages us to further explore the neural mechanisms underlying adaptive defensive behaviors. Once again, we are grateful for the kind words and supportive feedback.

Reviewer #3 (Recommendations for the authors):1. In this study, Pyeon et al. employ a predator-like robot to chase the animals to simulate naturalistic threat imminence in mice. While the idea and attempt are well intentioned, the overall data suggests the model is only inducing weak unconditional responses to the robot. Mice show only a small number of fleeing attempts compared to the number of robot chase trials.

We thank the reviewer for the valuable comments and agree that the flight count we initially presented may not fully capture the robustness of the behavior. To address this, we re-analyzed our flight data using the flight score method described by Fadok et al., which is calculated by dividing the average speed during each CS by the pre-CS speed over 10 seconds (Fadok et al., 2017). This method better reflects both the speed and the number of fleeing bouts during the CS.

Our re-analysis revealed that even mice in the control group exhibited higher flight scores during conditioning, with an average score of approximately 4. Additionally, using this more sensitive method of quantifying flight responses, we observed a significant difference in flight scores between the ChR2 and control groups during conditioning—a difference not evident with our previous counting method. The manuscript has been updated accordingly, including revised figures and additional velocity data during conditioning.

These updated results confirm that the robot paradigm induces stronger flight responses than previously reported, and the activation of CGRP neurons further amplifies these responses. We appreciate the reviewer’s valuable feedback, which allowed us to enhance our data and improve the manuscript.

2. For the conventional threat conditioning experiment, the authors subjected previously robot-conditioned mice to sequential conditioning. This resulted in a higher fleeing count in the ChR2 group during the habituation session (Figure 4 C). This effect could be due to a sensitization effect caused by previous ChR2 stimulation of the CGRP neurons.

We appreciate the reviewer for the thoughtful comment. The reviewer is correct that previously robot-conditioned mice exhibited a higher fleeing count in the ChR2 group during the habituation session and now also in the first trial block with our new analysis. This increased response could indeed be attributed to sensitization or fear generalization from the prior exposure to the robot threat. While we attempted to mitigate by allowing a one-week resting period and using a different tone for the CS, it is possible that some degree of sensitization or generalization remained.

However, we would like to emphasize that this heightened fleeing response was transient and did not persist throughout the conditioning trials. Importantly, during the retention test, the ChR2 group showed similarly low levels of flight scores compared to the control group. We appreciate the reviewer's valuable input and have carefully considered these observations. While sensitization may have influenced the early fleeing responses, we believe the overall conclusions of our study remain robust. The transient nature of this response does not detract from our finding that activation of CGRP neurons in the conventional threat conditioning paradigm primarily induces freezing behavior.

3. The speed of the robot used in this study (70-90 cm/s) is quite high and is likely to cause frequent collisions with the mouse. The paper does not describe what happens if the robot collides with the mouse. Does it stop? Does it push the mouse forward? The behavior during the conditioning session needs to be reported in much more detail. Also, the small number of flight bouts in all experiments suggests that the robot is not being perceived as a high-intensity threat.

We appreciate the reviewer’s concern regarding the robot's speed and its interactions with the animals. Given that this robot-based paradigm is relatively novel, we agree that the interactions between the robot and the animals, particularly during collisions, should have been explained in more detail.

The robot speed, ranging from 70 to 90 cm/s, was carefully selected after testing various speeds to ensure that it effectively induces a conditioned response without posing harm to the animals. When animals collided with the robot while running away, the robot would push them, causing the animals to increase their fleeing speed. In rare instances where an animal froze during the chase, the robot would push the animal forward; however, due to friction and reduced motor power, the robot's speed would naturally slow down, ensuring it does not run over or harm the animal but pushes it forward at a lower pace.

As the reviewer rightly pointed out, the behavior during the conditioning session requires more detailed reporting. In response, we have included velocity data for 10-s intervals – before, during, and after the CS – in Figure 3D and 5C. This data provides a more detailed visualization of the animals' movement patterns in relation to the robot’s speed during conditioning. By offering velocity data at these key time intervals, we believe it effectively captures and illustrates how the animals react to the robot's approach and collisions, providing a clearer representation of their behavior throughout the session. Additionally, we have updated the manuscript to include detailed information on what occurs during collisions between the robot and the animals. The updated section states:

“When animals collided with the robot, it pushed them forward, increasing their fleeing speed. If an animal blocked its path, the robot continued to push it, but its speed decreased due to friction and reduced motor power, ensuring it did not run over the animal.”

Regarding the small number of flight bouts in our experiments, our updated analysis now reveals significantly higher flight scores, indicating that the robot is perceived as a higher-intensity threat. The revised figures and analysis are reflected in the updated manuscript.

4. It is exceedingly difficult to discern how fleeing counts are defined and quantified. Because fleeing is being reported as counts, this indicates that these are discrete data, therefore, ANOVA is not an appropriate statistical method. ANOVAs assume continuous data and a normal distribution.

We appreciate the reviewer’s feedback regarding the quantification of fleeing counts. While we aimed to clarify this in the original manuscript, we recognize that further explanation may be needed. What we have done is extract the raw velocity data and count the peaks that exceeded 8 cm/s, with a minimum inter-peak interval of 0.6 s. The 8 cm/s threshold was determined by comparing the velocity data with experimenter observations of fleeing behavior, ensuring that all movements at or above this speed were consistently identified as fleeing events. However, in response to the reviewer’s concerns about the small number of flight counts, we transitioned to using flight scores rather than counts. We made sure that the method for quantifying the flight score is clearly explained in the manuscript. The revised section in the manuscript now states:

“For fleeing responses, flight score was calculated following previous studies (Fadok et al., 2017; Borkar et al., 2024). Speed (cm/s) was extracted using the animal’s center body point, and the flight score was measured by dividing the average speed during each CS alone period by the speed during the same length of the pre-CS period. For conditioning trials, the CS alone period was the first 7 s from CS onset, while for the habituation and retention tests, it was 10 s. Vertical movements, such as jumping, were manually recorded by an experimenter blind to the group assignments, with 1 point added to the flight score for each escape jump.”

We appreciate the reviewer’s thorough attention to the statistical analysis in our study. With the flight count data, we first confirmed that it met the normality assumptions and proceeded with ANOVA. However, we realize that we did not clearly mention this in the manuscript, which may have caused some confusion regarding the appropriateness of using ANOVA. We have updated all the statistics and figures in the manuscript accordingly. We sincerely thank the reviewer for this valuable opportunity to reanalyze the data, which has further strengthened the robustness of our findings. However, transitioning to flight scores have addressed this issue, as the flight scores allowed us to appropriately meet the assumptions required for ANOVA.

We have updated all the statistics and figures in the manuscript accordingly. We sincerely thank the reviewer for this valuable opportunity to reanalyze the data, which has further strengthened the robustness of our findings.

5. In Figure 3 B-C, ChR2 stimulation did not further elevate the freezing or fleeing count during the conditioning phase. Since US-mediated neuronal activation was elevated for at least 3 sec post termination of the US (Figure 1F, H), ChR2 stimulation should be performed for such an extended period. If possible, authors should also present the post-US data (do the mice freeze after the robot?).

We have updated all the statistics and figures in the manuscript accordingly. We sincerely thank the reviewer for this valuable opportunity to reanalyze the data, which has further strengthened the robustness of our findings. As the reviewer correctly pointed out, our electrophysiology data show that CGRP neurons remain elevated for at least 3 s following the termination of the US. While extending ChR2 stimulation for approximately 6 s could provide further insights, we found that even with the shorter 3-s stimulation, the ChR2 group showed a significantly elevated flight response in the updated analysis, which we believe effectively captures the impact of the stimulation.

Additionally, we have included velocity data during conditioning, showing the speed of animals during the pre-CS (10 s), CS, and post-CS (10 s). This data reveals a significant drop in speed during the post-CS period, indicating that the animals were most likely freezing after being chased by the robot. This freezing response was consistently observed across both groups, excluding the Jaws group.

6. The intensity of the pinprick and tail pinches are not defined and are potentially inconsistent across the mice as these stimuli are administered manually.

We agree with the reviewer that the manual administration of pinprick and tail pinches introduces potential inconsistencies across trials, which is indeed a limitation of our study. While footshocks could have provided more consistent stimuli, we opted against their use, as they can introduce noise and artifacts. Although it is challenging to precisely define the intensity of pinprick and tail pinch, we selected these aversive stimuli for their suitability to our experimental design. To minimize variability, a trained experimenter administered the stimuli throughout all recording sessions. We have included details about these efforts, along with the needle size used, in the Methods section.

7. On page 9, the authors state the following about the ChR2 stimulation, "Combined with the spontaneous firing rate during robot chasing (about 40 Hz), this totaled approximately 70 Hz, matching the peak firing rate observed in the upper quartile during tail pinch". ChR2 simulation will entrain the neuron to fire with each pulse up to approximately 20 Hz, after which the failure rate increases. In effect, neurophysiology does not work in an additive fashion.

What we intended to convey was that the perceived threat level was increased by additionally activating CGRP neurons with the spontaneous firing rate observed in our electrophysiology experiments, in conjunctions with the presentation of the robot or footshock US. However, we agree that the reviewer is correct in stating that neurophysiology does not work in an additive fashion, and we acknowledge that this point should be clarified in the manuscript. We appreciate the reviewer’s insightful comment and have now corrected this. The revised manuscript now states:

“To effectively enhance the general alarm signal, additional activation of CGRP neurons was applied at 30 Hz during the robot chasing.”

8. The authors claim, "We hypothesized that CGRP neurons contribute to the selection of adaptive defensive behaviors by signaling the severity of the threat". The selection of defensive behavior is not tested in these experiments

We appreciate the reviewer’s thoughtful comment. We acknowledge that our experiments did not directly test the selection process of defensive behaviors. Instead, our focus was on how CGRP neurons modulate defensive responses based on threat intensity. We have revised the manuscript to more accurately reflect the scope of our study and adjusted the phrasing to avoid any unintended overstatement. We have incorporated this change in the manuscript as follows:

“We hypothesized that CGRP neurons modulate adaptive defensive behaviors by signaling the intensity and type of threat.”

[Editors’ note: what follows is the authors’ response to the second round of review.]

The three reviewers agree that the revisions made strengthened the manuscript. However, additional revisions are needed. These are addressed in full below. The most important changes are clarifying language about the non-associative nature of flight and the role for CGRP neurons in flight vs US activity.Reviewer #1 (Recommendations for the authors):In my initial review I expressed concern that the flight observed to the cue was not due to pairing with the chasing robot, but would be observed to any cue played in the context. In response, the authors ran a cohort of unpaired mice which received cue presentations and the chasing robot at different times. I appreciate the authors including this control group. However, as was my concern, there is very poor evidence that flight in this setting is due to pairing with the chasing robot. Within the chasing sessions, paired and unpaired mice are showing comparable levels of flight. Further, in the extinction test unpaired mice appear to show even greater flight than the paired mice.

We appreciate the reviewer’s previous suggestion regarding the inclusion of an unpaired control group, which allowed us to more thoroughly evaluate the role of CS-US pairing in flight responses. In response to the reviewer’s current comments, we would like to address specific points regarding the flight scores observed in the paired and unpaired groups In particular, we would like to highlight that the unpaired group exhibited significantly lower flight scores compared to the control group, during the chasing sessions.

During the conditioning sessions, the unpaired control group actually exhibited significantly lower flight scores than the paired group (repeated-measures two-way ANOVA, F(3, 36) = 102.05, *p* = .000; post-hoc test, *p* = .018), suggesting that they failed to effectively form fear memory. This result supports the importance of CS-US pairing in driving conditioned flight behavior. For the retention test, while the reviewer correctly noted that the unpaired group exhibited higher flight scores than the paired control group, we would like to clarify that this difference is attributable to the methodology of our updated flight score metric, which calculates the ratio of velocity during the CS to velocity during the pre-CS. While this method provides a sensitive measure of flight responses, it is also susceptible to small variations in pre-CS velocity. For instance, when animals moved slightly more during the CS compared to the pre-CS, their flight score could appear inflated, even if their actual velocity during the CS was insufficient to qualify as true flight behavior.

In the unpaired group, although flight scores during retention test were higher, their overall velocity in response to the CS did not exceed 3 cm/s (Figure 3—figure supplement 1D). This velocity range is more consistent with exploratory behavior rather than flight. We recognize that this distinction may not have been sufficiently explained in the original manuscript, potentially leading to confusion. To address this, we have updated the manuscript as follows:

“The unpaired group exhibited significantly lower freezing levels compared to the paired control group. However, flight scores in the unpaired group were significantly higher than those in the control group. This was likely due to the tendency of the control group to remain frozen before and during the CS presentation. The unpaired group, however, showed sensory orientation responses to the CS, contributing to their elevated fleeing scores. Moreover, the movement speed of the unpaired group during the tone CS did not exceed 3 cm/s, a range typically associated with exploratory behavior in open field tests, suggesting exploratory rather than defensive behavior in the test environment (Figure 3−figure supplement 1D).”

By contrast, freezing shows specificity to paired mice and is readily observed during extinction testing. This is problematic for flight because it shows that the same mice not showing conditioned flight show conditioned freezing. Recent studies in both mice and rats that have used appropriate control cues have observed robust conditioned flight. That bar is not reached in this manuscript.

As we have previously stated, the unpaired group displayed significantly lower levels of flight during the conditioning session, supporting the specificity of flight responses to CS-US pairing. However, the reviewer is correct that freezing behavior shows clear specificity to the paired control group, and we acknowledge this point. However, we would like to emphasize the defensive behavior observed in the control animals when the robot speed was escalated. Under high-speed conditions (90 cm/s), conditioned flight responses were reliably observed in the control animals. We believe the 70 cm/s condition used in the main paradigm may not have been sufficient to induce robust flight responses in animals. Nevertheless, even under the 70 cm/s condition, flight responses were reliably observed across trials of conditioning.

To further address this concern, we have included velocity traces from a representative animal across 10 trials of the conditioning session in the revised manuscript (Figure 1—figure supplement 1C). These traces demonstrate how an individual animal engages in a range of defensive behaviors during the CS, including freezing, flight, and transitions between these behaviors. We hope that this additional data and explanation address the reviewer’s concern and demonstrate that our novel paradigm reliably observes robust conditioned flight responses, further supporting the validity of our findings.

Because the authors addressed my concerns, I am positively inclined towards the manuscript. However, because there is very little evidence that conditioned active defensive behaviors were observed, the authors need to refrain from using the term in the title and the manuscript. Further, the authors need to describe flight in a way that does not describe or suggest that it is conditioned or learned. This only seems correct given their observations of unpaired responding.

We thank the reviewer for their valuable feedback and for guiding us toward a better interpretation of our data. Including the unpaired control group has been instrumental in evaluating the specificity of conditioned flight responses and ensuring a robust analysis. We recognize that our initial explanation may not have sufficiently clarified the differences in flight responses between the paired and unpaired groups. Specifically, the unpaired group exhibited fewer flight responses compared to the control group during conditioning, and while their flight scores appeared higher during the retention test, this likely reflected exploratory behavior rather than true flight.

In response to the reviewer’s insightful comment, we have revised our manuscript to better explain these findings and to ensure the terminology used accurately reflects the observations. We hope these revisions provide greater clarity and further strengthen the manuscript.

Reviewer #3 (Recommendations for the authors):The manuscript by Pyeon et al. has been improved by the addition of additional analysis, an unpaired control condition, and clarification of experimental design. I have a few remaining concerns which should be addressed.

We appreciate the reviewer's previous comments, which have significantly strengthened our findings. We look forward to addressing the remaining concerns and making further updates to ensure the manuscript is as robust and clear as possible.

1. There remain numerous instances in the text that imply that CGRP neurons are capable of inducing escape behavior (Lines 59-60; 71-72; 260-262; 315-316), yet this is not demonstrated experimentally. In fact, when CGRP neurons are optogeneticly activated in the absence of external stimuli, they induce freezing behavior, not flight (Figure 2; Lines 142-143). What the authors show is that CGRP neurons are important for US encoding, which under the right experimental conditions can allow the CS to elicit flight behavior (i.e., under high threat conditions). The authors should change their language or demonstrate that CGRP neurons can elicit flight. Further to this point, the TetTox approach does not allow for the disambiguation of CGRP neuron involvement in different phases of learning or expression (Lines 312-313).

We completely agree with the point that activation of CGRP neurons does not directly induce active defensive behavior. To address this, we have carefully revised all sections of the manuscript to ensure that our language no longer implies that CGRP neuron activation induces or triggers active defensive behavior. Below are examples of how we revised the manuscript, including updates to the last paragraph of our Discussion section. We sincerely appreciate the reviewer’s insightful comment, which has greatly clarified the interpretation of our findings.

“However, for CGRP neurons to serve as true general alarm system, they must be capable of transmitting threat-related signals and facilitating the coordination of appropriate defensive behaviors, whether passive or active.

These results highlight the role of CGRP neurons as a general alarm signal, primarily facilitating passive defensive behaviors, while also engaging in active defensive behaviors in response to high-threat conditions.

In conclusion, by employing both conventional footshock and a naturalistic paradigm, the present study emphasizes the role of CGRP neurons in facilitating both passive and active defensive behaviors. Optogenetic stimulation of CGRP neurons in the absence of external stimuli induced robust freezing, and their activation during conventional fear conditioning further amplified conditioned freezing, demonstrating their primary role in driving passive defensive responses. However, under heightened threat conditions, such as enhanced CGRP activation or faster robot speeds, these neurons also strengthened active defensive behaviors by amplifying perceived threat. These findings suggest that CGRP neurons detect and process threats, predominantly driving freezing behavior, while also enabling active responses under heightened danger to facilitate appropriate defensive behaviors aligned with the intensity of the threat.”

2. The authors describe what happens when the robot collides with the mouse (e.g., Lines 89-90). The authors should provide a video demonstrating this, and I would like to see a quantification of how the animal's speed is altered by collision. The authors also state that bumping leads to stronger conditioned defensive behaviors (Lines 155-156) but these data are not shown. This correlation should be shown.

We appreciate the reviewer’s attention to the details of our paradigm. In the previously included video (Movie 1), we demonstrated an animal’s response to the CS during conditioning and its interaction with the chasing robot. While this video already depicted changes in the animal's movement following collisions, our initial focus was on highlighting the flight response to the CS in the ChR2 group. However, as the reviewer pointed out, given that this is a novel paradigm, we recognize the value of providing a more detailed understanding of how the animal's movements are influenced by the robot. To address this, we have now referenced this video in the bumping section to provide additional context and enhance the interpretation of the paradigm.

Additionally, in response to the reviewer's comment, we have included a velocity trace from a representative animal to illustrate how speed changes after a collision (Figure 3—figure supplement 1C). In this figure, collision incidents are indicated with red arrows, demonstrating that animals tend to move forward with increased speed following a collision with the robot.

**Author response image 1. sa2fig1:** 

Regarding our earlier statement that bumping leads to stronger conditioned defensive behaviors, our intention was to emphasize that there were no group differences in bumping incidents across the four groups. In response to the reviewer's comment, we conducted additional analyses to examine the correlation between bumping incidents and conditioned defensive behaviors, as illustrated in Author response image 1. While we did not find statistically significant correlations between bumping and defensive behaviors in most groups, we observed a positive correlation between bumping and freezing in the ChR2 group (r = 0.65, *p* < 0.05). Based on these findings, we have removed the phrase 'bumping leads to stronger conditioned defensive behaviors' from the manuscript to avoid any potential misinterpretation. Therefore, we have revised the description to clarify our intention, and the updated manuscript now states:“Physical bumping occurred during robot chasing, potentially influencing the perception of threat. To ensure that differences in defensive behavior were not due to variations in bumping among the four groups, we analyzed bumping incidents and found no significant differences (Figure 3−figure supplement 1C; Movie 1). This confirms that any observed differences in defensive responses are likely attributable to alterations in CGRP neuronal activity.”

3. The argument given in Lines 174-176 should be rephrased to indicate that this is a hypothesis for why the Jaws group had freezing similar to control levels.

We appreciate the reviewer’s insightful comment, which allowed us to clarify our hypothesis regarding why the Jaws group exhibited freezing levels similar to the control group. Based on a published study in *Nature Neuroscience*, the use of Jaws, a red-shifted rhodopsin, can result in rebound activation even with ramped illumination (Chuong et al., 2014), as employed in our experiment. This rebound activation after the laser is turned off may explain the freezing behavior observed in the Jaws group. We have rephrased this section in the manuscript to clearly present this as a hypothesis and to better support our interpretation with existing literature. The revised text now states:

(Result section) “This result may be attributed to post-illumination rebound excitation, as Jaws has been shown to yield residual activity even when ramped illumination is used to minimize this effect (Chuong et al., 2014).”

(Discussion section) “However, the reduced fear responses were not sustained on the retention test day, acquisition of fear memory (Figure 3E). It has been reported that square pulse can cause strong rebound excitation following inhibition (Chuong et al., 2014). While ramped illumination reduces the magnitude of rebound excitation, it does not eliminate it entirely, leaving small residual excitation. This residual activity may have contributed to the formation of fear memory despite the inhibition of CGRP neurons.”

4. The authors claim that collisions with the robot push the mouse forward but do not harm the animal, yet in Lines 298-299 they posit that prolonged neuronal excitation was "presumably due to the pain experienced after bumping into robot". Which is it, painful or innocuous?

We appreciate the reviewer’s thoughtful comment highlighting potential confusion in our language. What we meant by “did not harm the animal” when selecting higher robot speeds was that we ensured the robot's speed was not so high as to physically run over or cause harm to the animals. However, we acknowledge that “do not harm” may unintentionally imply an absence of pain. While we cannot directly measure pain in the animals, it is possible that chasing at high speeds and collisions could cause some discomfort or distress.

However, based on the reviewer’s previous comment regarding our hypothesis about the Jaws group's inhibition of CGRP neurons being insufficient, this section has now been removed. We are grateful for the reviewer’s attention to this matter and will take extra care to ensure clarity and precision in our terminology.

5. A more accurate Figure title for Figure 3 would be "Activation of CGRP neurons during US presentation enhances active defensive behavior".

We sincerely thank the reviewer for their thoughtful and precise suggestion. We agree that the title "Activation of CGRP neurons during US presentation enhances active defensive behavior" more accurately reflects the content of Figure 3. Accordingly, we have updated the figure title in the revised manuscript.

6. Figure 5 needs a new title as CGRP neurons are not manipulated or recorded in these experiments.

The reviewer is correct that CGRP neurons are not manipulated or recorded in Figure 5. We have updated the title to: "Escalating threat intensity modulates defensive behavior" to more accurately reflect the content of the figure.

7. The authors should comment on the fact that the TetTox mice react normally to US exposure (Figure 6). Which pathways are mediating the US behavioral response?

As the reviewer noted, when CGRP neurons are silenced, responses to the US are generally reduced. For instance, in footshock experiments, silencing CGRP neurons impairs US processing, resulting in diminished sensitivity to the shock. However, during robot chasing, the mechanical properties of the US physically push the animals forward. This mechanical force is independent of CGRP neuron activity and compels the animals to move, even though their defensive responses to the CS are reduced compared to controls. As a result, the velocity of TetTox mice during the US appears similar to that of control animals, as their movement is driven by the physical force of the robot rather than by fear processing. However, information related to the US can still be processed through alternative mechanisms independent of CGRP neurons, as evidenced by the progressive increase in freezing levels observed in TetTox mice during conditioning under high-threat conditions with a robot speed of 90 cm/s (Figure 6B).

Reference

Chuong, A. S., Miri, M. L., Busskamp, V., Matthews, G. A., Acker, L. C., Sørensen, A. T., Young, A., Klapoetke, N. C., Henninger, M. A., and Kodandaramaiah, S. B. (2014). Noninvasive optical inhibition with a red-shifted microbial rhodopsin. *Nature neuroscience*, *17*(8), 1123-1129.